# RECURRENT LINEAR TRANSFORMERS

## ABSTRACT

The self-attention mechanism in the transformer architecture is capable of capturing long-range dependencies and it is the main reason behind its effectiveness in processing sequential data. Nevertheless, despite their success, transformers have two significant drawbacks that still limit their broader applicability: (1) In order to remember past information, the self-attention mechanism requires access to the whole history to be provided as context. (2) The inference cost in transformers is expensive. In this paper we introduce recurrent alternatives to the transformer self-attention mechanism that offer a context-independent inference cost, leverage long-range dependencies effectively, and perform well in practice. We evaluate our approaches in reinforcement learning problems where the aforementioned computational limitations make the application of transformers nearly infeasible. We quantify the impact of the different components of our architecture in a diagnostic environment and assess performance gains in 2D and 3D pixel-based partially-observable environments. When compared to a state-of-the-art architecture, GTrXL, inference in our approach is at least 40% cheaper while reducing memory use in more than 50%. Our approach either performs similarly or better than GTrXL, improving more than 37% upon GTrXL performance on harder tasks.

## 1 INTRODUCTION

Transformers (Vaswani et al., 2017) have achieved state-of-the-art performance in many sequential data processing problems, such as natural language processing (e.g., Brown et al., 2020; Devlin et al., 2018) and computer vision (e.g., Petit et al., 2021; Zhong et al., 2020). These successes are often attributed to the transformers self-attention mechanism which can to capture long-range dependencies. Typically, the self-attention mechanism operates on the whole sequence at once and it uses a dot product coupled with a softmax function to extract relationships between elements in the sequence.

Despite empirical success, transformers have two main limitations: (1) the context length limits how far back in the sequence the transformer can model, and (2) its inference cost—the computational cost of applying self-attention to a single element in the sequence—is high compared with alternatives like recurrent neural networks. In fact, these issues are coupled because increasing the context length in a transformer architecture leads to even higher inference costs. Addressing these issues is now a major research topic (e.g., Dai et al., 2019; Choromanski et al., 2020; Bulatov et al., 2022).

The Linear Transformer architecture is an approach designed to reduce the computational complexity of the self-attention mechanism (Katharopoulos et al., 2020). This approach uses a generic kernel function instead of the softmax, what allows it to be updated iteratively instead of requiring the entire context. Unfortunately, this approach has three main limitations: (1) its self-attention mechanism naively adds positive values to the recurrent state, which can lead to instability when processing long sequences due to continual growth. (2) Performance is dependent on the choice of the kernel function—the element-wise feature maps used in the original paper, for example, have been shown to have limited memory capacity (Schlag et al., 2021). Lastly, (3) the Linear Transformer's self-attention mechanism maintains a matrix as a recurrent state, which can result in a high memory cost when multiple self-attention heads are used.

In this paper we introduce two recurrent alternatives extending the Linear Transformer's self-attention mechanism and that address the issues aforementioned. Our first contribution, Recurrent Linear Transformer (ReLiT), uses a gated structure that allows it to uncover relationships far in the

past. It also uses a different self-attention mechanism that can *learn* a highly parallelizable feature map that is amenable to sequential computation with a context-independent inference cost. Our second contribution, Approximate Recurrent Linear Transformer (AReLiT), introduces an approximate version of ReLiT's self-attention, eliminating the need to maintain a matrix as a recurrent state.

We evaluate the proposed approaches in reinforcement learning (RL) problems, where reducing computation and memory are key to enable transformers-based agents to learn while interacting with the world. A slow inference step reduces how quickly the agent can update and select new actions, dramatically increasing runtimes or negatively impacting performance in real-time environments. In addition, contexts large enough to produce good performance are often not practical. Many RL problems are partially observable and it is not feasible for the agent to store a long history of interaction. Even simple RL problems require hundreds of millions of interactions and episodes over 100,000 steps long (Nair et al., 2015; Machado et al., 2018). These numbers are already much larger than what most transformer systems can process. These characteristics make it difficult to apply current methods, even the linear transformer, to online RL.

Concretely, we first investigate our architecture in the T-Maze environment (Bakker, 2001): a small diagnostic environment designed to test an agent's ability to remember information far in the past. We show that limiting the input context of the canonical self-attention mechanism has a detrimental effect on performance and that a large input context, albeit at the cost of increased computational complexity, is necessary for this task. Both ReLiT and AReLiT match the performance of much more computationally expensive transformer architectures. We then extend these results to the larger Mystery Path problem (Pleines et al., 2023), which is pixel-based navigation task that requires the agent to memorize a long sequence of steps. In Mystery Path, our approach outperforms the state-of-the-art transformer architecture in reinforcement learning, GTrXL (Parisotto et al., 2020), by more than 37%. Finally, we extend these results to the larger Memory Maze (Pašukonis et al., 2023) problem, illustrating that the performance of AReLiT is close to GTrXL while reducing computation and memory 40% and 50% respectively.

## 2 PRELIMINARIES

In this section, we provide a brief overview of what is required to understand our proposed transformer approach. We first discuss the canonical transformer architecture and then we discuss the Linear Transformer approach, which is the basis of our approach.

### 2.1 CANONICAL TRANSFORMER ARCHITECTURE

The Transformer architecture was introduced for supervised next token prediction tasks (Vaswani et al., 2017). Our main contribution is a new self-attention mechanism; this section provides the background required to understand the self-attention mechanism in transformers.

Self-attention is mechanically simple. For a given query token $i$ (embedded in $\mathbf{x}_i \doteq \mathbf{X}(\mathbf{i}, \cdot)$), we output an embedded context vector that weights each input token's importance (attention weighted) to the query token. The input to the self-attention layer is a matrix $\mathbf{X} \in \mathbb{R}^{N \times d}$, an embedding of each input token (1 to $N$) into a vector, $\mathbb{R}^d$. The output is a matrix $\mathbf{A} \in \mathbb{R}^{N \times d_h}$, where $d_h$ is the head dimension. Algorithm 1 shows a single self-attention layer with learnable parameters $\mathbf{W}_Q, \mathbf{W}_K, \mathbf{W}_V \in \mathbb{R}^{d \times d_h}$.

---
**Algorithm 1** Canonical Self-Attention

**Input**: $\mathbf{X} \in \mathbb{R}^{N \times d}$
**Parameters**: $\mathbf{W}_Q, \mathbf{W}_K, \mathbf{W}_V \in \mathbb{R}^{d \times d_h}$

1: $\mathbf{Q} \leftarrow \mathbf{X}\mathbf{W}_Q$
2: $\mathbf{K} \leftarrow \mathbf{X}\mathbf{W}_K$
3: $\mathbf{V} \leftarrow \mathbf{X}\mathbf{W}_V$
4: $\mathbf{A} \leftarrow softmax(\frac{\mathbf{Q}\mathbf{K}^\intercal}{\sqrt{d}})\mathbf{V}$

**Output**: $\mathbf{A} \in \mathbb{R}^{N \times d_h}$

---

We can think of the process in two steps. In step one we calculate the attention weights. We compare each token in the context to all other tokens in the context ($\boldsymbol{Q}\boldsymbol{K}^T$). The weights are then scaled the size of the embedding dimension and normalized with an element-wise *softmax*. In step two, we compute and return the attention-weighted context vectors, one for each input in $\mathbf{X}$.

The self-attention mechanism in Algorithm 1 is computationally expensive. The inference cost of self-attention, the cost for processing a single element in a sequence, depends on the input sequence length $N$. For a naive implementation, the inference cost has $\mathcal{O}(Nd^2)$ time and $\mathcal{O}(Nd)$ space

complexity; increasing the sequence length quadratically increases the computational complexity. A simple mitigation is to limit the size of the input sequence by maintaining a window of the history of input activations in memory (Dai et al., 2019), but doing so limits the past information the self-attention mechanism can recall.

## 2.2 RECURRENT ATTENTION WITH LINEAR TRANSFORMERS

The Linear Transformer architecture (Katharopoulos et al., 2020) introduces a general way of formulating self-attention as a recurrent neural network by replacing the softmax with a kernel function, leveraging its equivalence to applying kernel smoothing over inputs (see work by Tsai et al., 2019).

A single time-step of inference of the Linear Transformer self-attention is described in Algorithm 2. Let $k(\mathbf{a}, \mathbf{b}) = \phi(\mathbf{a})^\intercal \phi(\mathbf{b})$, where $\phi : \mathbb{R}^{d_h} \to \mathbb{R}^{d_k}$ is a non-linear feature map, $d_k$ is the output dimension of the feature map $\phi$, and $k : \mathbb{R}^{d_h} \times \mathbb{R}^{d_h} \to \mathbb{R}^+$. Additionally, let $\otimes$ be defined as the vector outer product operation. At a given timestep $t$, the Linear Transformer self-attention maintains a matrix $\mathbf{C}_{t-1} \in \mathbb{R}^{d_h \times d_k}$ and a vector $\mathbf{s}_t \in \mathbb{R}^{d_k}$ as a recurrent state, which is updated iteratively using the current input vector $\mathbf{x}_t$. Different from Algorithm 1, Algorithm 2 applies the feature map $\phi$ to generate the query and key for a given time-step (lines 1 and 2). The Linear Trans-

> **Algorithm 2** Linear Transformer's Self-Attention
> ___
> **Input**: $\mathbf{x}_t \in \mathbb{R}^d$, $\mathbf{C}_{t-1} \in \mathbb{R}^{d_h \times d_k}$, $\mathbf{s}_{t-1} \in \mathbb{R}^{d_k}$
> **Parameters** : $\mathbf{W}_Q, \mathbf{W}_K, \mathbf{W}_V \in \mathbb{R}^{d_h \times d}$
> $\mathbf{s}_0 \leftarrow \mathbf{0}, \mathbf{C}_0 \leftarrow \mathbf{0}$.
>
> 1: $\mathbf{q}_t \leftarrow \phi(\mathbf{W}_Q \mathbf{x}_t)$
> 2: $\mathbf{k}_t \leftarrow \phi(\mathbf{W}_K \mathbf{x}_t)$
> 3: $\mathbf{v}_t \leftarrow \mathbf{W}_V \mathbf{x}_t$
>
> 4: $\mathbf{C}_t \leftarrow \mathbf{C}_{t-1} + \mathbf{v}_t \otimes \mathbf{k}_t$
> 5: $\mathbf{s}_t \leftarrow \mathbf{s}_{t-1} + \mathbf{k}_t$
>
> 6: $\mathbf{a}_t \leftarrow (\mathbf{C}_t \mathbf{q}_t)/(\mathbf{s}_t^\top \mathbf{q}_t)$
>
> **Output**: $\mathbf{a}_t \in \mathbb{R}^{d_h}$, $\mathbf{C}_t \in \mathbb{R}^{d_h \times d_k}$, $\mathbf{s}_t \in \mathbb{R}^{d_k}$

former self-attention stores the outer product of value and key vectors as a recurrent state matrix $\mathbf{C}_t$ (line 4). Additionally, the sum of the key vectors is stored as a recurrent normalization vector $\mathbf{s}_t$ (line 5). The attention output vector, $\mathbf{a}_t$, is calculated by multiplying the recurrent state with the query vector, and normalizing it using the product of the normalization vector, $\mathbf{s}_t$, and the query vector, $\mathbf{q}_t$ (line 6).

The Linear Transformer's self-attention has a context-independent inference cost, unlike the canonical self-attention mechanism. In Algorithm 2, processing a single input vector ($\mathbf{x}_t$) has a space and time complexity of $\mathcal{O}(dd_k)$, assuming $d$, the embedding dimension (of the input), is greater than $d_h$, which is the size of the attention-weighted context vector $\mathbf{a}_t$. Unlike vanilla self-attention, the computational complexity does not depend on the context length, making it more efficient for longer sequences.

## 3 RECURRENT LINEAR TRANSFORMERS (ReLiT)

In this section we introduce ReLiT, to addresses two of the limitations of Linear Transformers. Specifically, (1) the recurrent equations in Algorithm 2 (lines 5 & 6) add positive values to the recurrent state, which could lead to potentially large recurrent states. (2) Performance critically depends on the choice of the kernel feature map $\phi$ (lines 1 & 2); element-wise functions such as the Exponential Linear Unit (ELU) typically perform worse than softmax (Katharopoulos et al., 2020).

ReLiT mitigates these two issues by introducing a gating mechanism and a parameterized feature map. The gating mechanism controls the flow of information at each index of $\mathbf{C}$ (the location of the recurrent states of the self-attention mechanism), allowing arbitrary context memory (inducing a trade-off with precision). The parameterized feature map is used to calculate the key and query vectors in the self-attention mechanism, eliminating the choice of kernel feature map $\phi$.

### 3.1 GATING MECHANISM TO CONTROL THE FLOW OF INFORMATION

In the Linear Transformer self-attention, at a given time-step $t$, Algorithm 2 increments the recurrent state, $\mathbf{C}_{t-1}$, and normalization vector, $\mathbf{s}_{t-1}$, (lines 4 and 5). Assuming $\mathbf{C}_0$ and $\mathbf{s}_0$ are initialized to zero, recall the update equations for $\mathbf{C}_t$ and $\mathbf{s}_t$ are recursively defined as follows:

$$\mathbf{C}_t \doteq \mathbf{C}_{t-1} + \mathbf{v}_t \otimes \mathbf{k}_t, \quad (1) \qquad\qquad \mathbf{s}_t \doteq \mathbf{s}_{t-1} + \mathbf{k}_t. \quad (2)$$

Equations 1 and 2 add arbitrary positive values to $\mathbf{C}_{t-1}$ and $\mathbf{s}_{t-1}$ (due to the positive feature map $\phi$) and have no way to control the flow of past information. The recurrent states could grow arbitrarily large, making prediction unstable. Instead, we use a normalized exponential average—with element-wise learned decay parameters—which smoothly reduces the impact of past information.

Gating mechanisms can be used to control the flow of information in recurrent updates. We propose a learned outer-product-based gating mechanism that decays every element of $\mathbf{C}_{t-1}$ and $\mathbf{s}_{t-1}$ allowing the network to learn the decay for each element (aka memory location). We introduce learnable parameters $\mathbf{W}_\beta \in \mathbb{R}^{d_h \times d}$, $\mathbf{W}_\gamma \in \mathbb{R}^{d_k \times d}$, and gating vectors $\beta_t$, and $\gamma_t$. Let $\sigma_g$ be a sigmoid function defined as $\sigma_g(x) \doteq \frac{1}{1+e^{-x}}$, we define $\beta_t$ and $\gamma_t$ as follows:

$$\beta_t \doteq \sigma_g(\mathbf{W}_\beta \mathbf{x}_t), \quad (3) \qquad\qquad \gamma_t \doteq \sigma_g(\mathbf{W}_\gamma \mathbf{x}_t). \quad (4)$$

Let $\odot$ be the element-wise product, we use the outer product of $\beta_t$ and $\gamma_t$ to control the flow of past information in recurrent states $\mathbf{C}_t$ and $\mathbf{s}_t$, modifying Equations 1 and 2 as follows:

$$\mathbf{C}_t \doteq \big((1 - \beta_t) \otimes (1 - \gamma_t)\big) \odot \mathbf{C}_{t-1} + \big(\beta_t \odot \mathbf{v}_t\big) \otimes \big(\gamma_t \odot \mathbf{k}_t\big), \quad (5)$$

$$\mathbf{s}_t \doteq (1 - \gamma_t) \odot \mathbf{s}_{t-1} + \gamma_t \odot \mathbf{k}_t. \quad (6)$$

We use outer products to learn the decay rate for each index of $\mathbf{C}_t$, without requiring individual parameters for each index. The outer product assumes the decay rate at each index is independent.

## 3.2 LEARNABLE FEATURE MAP FOR SELF-ATTENTION

Recall that the self-attention mechanism of the Linear Transformer uses a kernel feature map to calculate the key and query vectors:

$$\mathbf{k}_t \doteq \phi(\mathbf{W}_K \mathbf{x}_t), \quad (7) \qquad\qquad \mathbf{q}_t \doteq \phi(\mathbf{W}_Q \mathbf{x}_t). \quad (8)$$

We consider a deterministic approach to learn the key and value vectors in the Linear Transformer self-attention mechanism. We introduce modifications to $\mathbf{k}_t$, $\mathbf{q}_t$, and gating vectors calculation described in Equations 7, 8, 3, and 4 respectively. We start by introducing a hyperparameter $\eta$ that controls the dimension of the feature maps used to construct the $\mathbf{k}_t$ and $\mathbf{q}_t$. Let $\mathbf{W}_{p_1}, \mathbf{W}_{p_2}, \mathbf{W}_{p_3} \in \mathbb{R}^{\eta \times d}$ be learnable parameters. We modify the dimensions of $\mathbf{W}_\gamma$ as $\mathbf{W}_\gamma \in \mathbb{R}^{d_h \times d}$, getting rid of $d_k$, the kernel feature map dimension. Let $\textit{flatten}()$ be a function that flattens a matrix into a vector. We redefine $\mathbf{k}_t$ and $\mathbf{q}_t$ (previously defined in Equations 7 and 8) as follows:

$$\mathbf{k}_t \doteq \textit{flatten}(relu(\mathbf{W}_{p_1}\mathbf{x}_t) \otimes relu(\mathbf{W}_K\mathbf{x}_t)) \quad (9) \qquad \mathbf{q}_t \doteq \textit{flatten}(relu(\mathbf{W}_{p_2}\mathbf{x}_t) \otimes relu(\mathbf{W}_Q\mathbf{x}_t)) \quad (10)$$

We also modify the gating vectors $\gamma_t$ calculation in Equation 4 as follows:

$$\gamma_t \doteq \textit{flatten}(\sigma_g(\mathbf{W}_{p_3}\mathbf{x}_t) \otimes \sigma_g(\mathbf{W}_\gamma\mathbf{x}_t)) \quad (11)$$

Using the modified key, query, and gating vectors, the recurrent states $\mathbf{C}_t \in \mathbb{R}^{d_h \times \eta d_h}$ and $\mathbf{s}_t \in \mathbb{R}^{\eta d_h}$ are calculated according to Equations 5 and 6. It is important to note that the feature map dimension, $d_k = \eta d_h$, is now controlled by the hyperparameter $\eta$. Equations 9 and 10 use outer products to learn multiplicative interactions in the key and query vectors. Learning multiplicative interactions in the feature vectors allows learning complex non-linear relationships through training instead of relying on an explicit non-linear element-wise function or on random feature maps.

Finally, we use the relu activation function to ensure the output of the feature map is positive. A positive feature map output is necessary as it ensures that the similarity scores produced by the underlying kernel function are positive.

The **Recurrent Linear Transformer** (ReLiT) architecture incorporates the changes discussed above into the Linear Transformer self-attention. The pseudo-code for ReLiT is available in Appendix B. ReLiT has similar space and time complexity as the Linear Transformer. For processing a single element in a sequence, ReLiT has a space and time complexity of $\mathcal{O}\left(\eta d^2\right)$ and $\mathcal{O}\left(\eta d^2\right)$, respectively. In comparison, Linear Transformer requires $\mathcal{O}\left(d_k d\right)$, and $\mathcal{O}\left(d_k d\right)$. Notice $d_k$ is defined to be the output dimension of the kernel feature map, which is $\eta d_h$ in ReLiT. Similar to Linear Transformer, the space and time complexity of ReLiT is independent of $N$ and only depend on static hyperparameters $d$ and $\eta$.

# 4 APPROXIMATE RECURRENT LINEAR TRANSFORMER (AReLiT)

Operating on large matrices is expensive. Recall that ReLiT stores a matrix of dimension $d_h^2 \eta$ as a recurrent hidden state. This becomes more problematic with the use of multiple heads and layers; which are typically required to improve stability during the training (see Michel et al., 2019). For example, state-of-the-art architectures use 8 heads and 12 layers; 96 heads in total (Parisotto et al., 2020). Second, the update to $\mathbf{C}_t$ makes use of expensive and memory heavy operations: an outer product, element-wise matrix sum, and multiplication.

Our second approach, called Approximate Recurrent Linear Transformer (AReLiT), uses a low-rank approximation to reduce the space complexity of ReLiT. We replace the previous recurrent state matrix $\mathbf{C}_{t-1}$ with a set of vectors, reducing the space complexity of ReLiT by $d$. We introduce an approximation of the Kronecker delta function using a sum of cosine functions and we use this to approximate $\mathbf{C}_{t-1}$.

Our goal is to approximate the recurrent state update in Equation 5 with an approximation that uses less space than $\mathcal{O}(\eta d^2)$. Recall that Equation 5 replaces $\mathbf{C}_t$ with $\mathbf{C}_{t-1}$ plus a new outer product. To derive an approximation, we want to replace $\mathbf{C}_{t-1}$ with a matrix that has a lower rank. Also, we want to derive an update rule that is an approximation of Equation 5, but instead of updating the full-rank matrix $\mathbf{C}_{t-1}$, we update the low-rank approximation.

We can use a sum of cosine functions to approximate a sum of outer products. This approximation is deterministic and does not introduce variance in the approximation, and it keeps incremental updates to the state end-to-end differentiable. Our approach is inspired by the rank-1 approximation introduced by Ollivier et al. (2015), but instead of using random numbers to approximate a Kronecker delta function, we use a trigonometric identity that relates a Kronecker delta function to an integral over cosines. Recall that the Kronecker delta function is defined for integers $m$ and $n$ such that $\delta_{mn} = 1$ if $m = n$, and $\delta_{mn} = 0$ if $m \neq n$. We present an approximation $\hat{\delta}_{mn}$ of $\delta_{mn}$ such that $\hat{\delta}_{mn}$ is defined as follows:

$$\hat{\delta}_{mn} \doteq \frac{2}{r} \sum_{i=0}^{r} \left( \cos\left(\frac{2\pi i}{r} m\right) \cos\left(\frac{2\pi i}{r} n\right) \right). \tag{12}$$

It can further be shown that $\lim_{r \to \infty} \hat{\delta}_{mn} = \delta_{mn}$. The derivation for this result is presented in Appendix C.1. We use the approximation of the Kronecker delta function in Equation 12 to approximate the recurrent state update in Equation 5. For a given $r$, we maintain recurrent states $\tilde{\mathbf{v}}_{t-1}^k$ and $\tilde{\mathbf{k}}_{t-1}^k$ for $k = 0, 1, \ldots, r$. For $\omega_k \doteq \frac{2\pi k}{r}$, and assuming $\tilde{\mathbf{v}}_0^i$ and $\tilde{\mathbf{k}}_0^i$ are initialized as zeros, we directly calculate the attention output, $\mathbf{a}_t$, in replacement of $\mathbf{C}_t$, considering the recurrent updates to $\tilde{\mathbf{v}}_t^i$ and $\tilde{\mathbf{k}}_t^i$:

$$\tilde{\mathbf{v}}_t^k \doteq \cos(\omega_k t)\beta_t \odot \mathbf{v}_t + (1 - \beta_t) \odot \tilde{\mathbf{v}}_{t-1}^k, \tag{13} \qquad \tilde{\mathbf{k}}_t^k \doteq \cos(\omega_k t)\gamma_t \odot \mathbf{k}_t + (1 - \gamma_t) \odot \tilde{\mathbf{k}}_{t-1}^k, \tag{14}$$

$$\mathbf{a}_t \doteq \frac{\sum_{k=0}^{r} \tilde{\mathbf{v}}_t^k \left( \left(\tilde{\mathbf{k}}_t^k\right)^\mathsf{T} \mathbf{q}_t \right)}{2r(\mathbf{s}_t^\mathsf{T} \mathbf{q}_t)}. \tag{15}$$

Due to space constraints, the rationale behind these approximations is presented in Appendix C.1.1. The pseudocode for AReLiT can also be found in the Appendix D. Unlike Equation 5, Equations 13 and 14 define a recurrence over vectors instead of matrices. If $r \ll d$, then the recurrence is more efficient in space than the recurrence in Equation 5. In Appendix E, we provide an empirical evaluation of the impact of different values of $r$: in the quality of the approximation, showing that, in practice, it seems small $r$ does not compromise the quality of the approximation or overall performance. The computational complexity of AReLiT is $\mathcal{O}(r\eta d)$ and $\mathcal{O}(d^2 + r\eta d)$ in space and time. With AReLiT, we have significantly improved the complexity of self-attention and these differences manifest in experiments as we show next. We compare the computational complexities of our proposed approaches to GTrXL Parisotto et al. (2020) in Appendix A.

# 5 EMPIRICAL EVALUATION

This section investigates our proposed approaches in several partially observable reinforcement learning (RL) control problems. As previously mentioned, we evaluate the architectures we in-

troduced in RL problems because this is a setting that is particularly challenging to transformers. In the RL, we need fast inference because of the interactive nature of the problem, and the agent might need to remember events far in the past. RL problems highlight these requirements more than most other benchmarks. The memory requirements vary across the environments we consider. In T-maze (Bakker, 2001), the agent must remember a single cue signal. In CartPole, the agent must estimate the hidden state by integrating information over time. In Mystery Path (Pleines et al., 2023), the agent must remember multiple locations in a grid environment. Finally, we also experiment with the Memory Maze environment (Pašukonis et al., 2023), which requires retaining the layout of a 3D maze in addition to several locations across the maze.

**Diagnostic MDP** The T-Maze environment is used to evaluate an agent's ability to learn long context dependencies in a reinforcement learning scenario (Bakker, 2001). In this environment, the agent must remember a cue shown only at the beginning of an episode in order to decide which way to turn at the end of a hallway (inset plot in Figure 1). The cue is only included in the observation on the first timestep. The difficulty of this environment can be increased by increasing the corridor length. The agent's actions are NSEW, and the observation is a binary encoding of the current cell (gray code), the cue (on the first step), and several random distractor bits. The full details are provided in Appendix G.1.

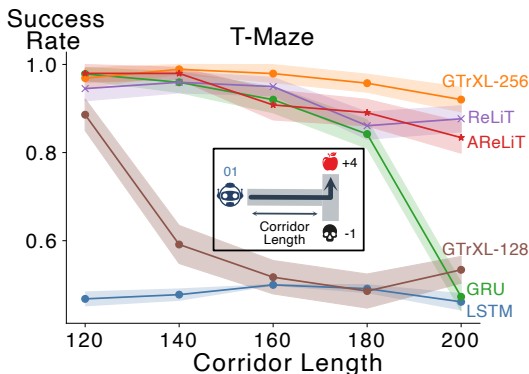

Figure 1: Success rate in the last 100K timesteps averaged over $50$ runs in T-Maze (shown inset). The shaded region represents the standard error.

We trained six agents for five million steps in the T-Maze environment, for corridor lengths 120–200. The network architecture for each agent has a shared representation learning layer, either an RNN or a transformer, which is then followed by separate actor and critic heads. Two of these agents were trained using an RNN as the shared representation layer, namely LSTM (Hochreiter & Schmidhuber, 1997) and GRU (Graves et al., 2016). The other two agents used a transformer, particularly the GTrXL architecture (Parisotto et al., 2020). In GTrXL, the memory size hyperparameter, defined as the amount of stored history, controls the context length. We train two GTrXL agents, GTrXL-128 and GTrXL-256, corresponding to memory sizes 128 and 256. Note that for the corridor lengths considered, GTrXL-256 has the entire episode provided as input. We also evaluate ReLiT ($\eta = 4$) and AReLiT ($\eta = 4, r = 1$); we do so by replacing the XL-attention of GTrXL with one of the two approaches, while preserving the order of the layers and the gating of GTrXL. This allows us to evaluate exactly the impact of the newly introduced self-attention mechanisms without other confounders. The base RL algorithm for all agents use Advantage Actor-Critic (A2C) (Wu et al., 2017). Architecture-specific hyperparameters and tuning strategies are described in Appendix G.1.

Figure 1 summarizes the main results. We report the success rate, the percentage of correct decisions, averaged over the last 100K timesteps of the experiment. An agent that chooses randomly at the intersection would achieve a success rate of $0.5$. In this experiment, GTrXL is sensitive to the amount of history provided as input; GTrXL-128 (brown) fails for corridor lengths greater than 120, whereas GTrXL-256 (orange) works well across all corridor lengths. ReLiT (purple) and AReLiT (red) match the performance of GTrXL-256 despite not having access to the entire episode as input. Note that AReLiT performs close to ReLiT even with $r = 1$ (the approximation parameter). GRU (green) outperforms LSTM (blue), but its performance drops in the longest corridor lengths.

We explored several ablations of our approach in the T-Maze, finding: (1) learning decay parameters for each element of $\mathbf{C}$ (gating) is better than a scalar decay used in the Linear Transformer (Peng et al., 2021), (2) our expansive feature map outperforms element-wise maps like ELU+1 and DPFP (Schlag et al., 2021), and (3) our low-rank sin-cos based approximation outperforms the rank-one approximation introduced by Ollivier et al. (2015). The results can be found in Appendix K.

AReLiT is more computationally efficient than GTrXL-256 in T-Maze. For a single attention head, AReLiT uses roughly 125.1 times fewer operations than GTrXL-256, and 36.57 times less space.

**Partially Observable Classic Control** We explored a two variants of CartPole (Barto et al., 1983), inspired by previous work (Morad et al., 2022; Duan et al., 2016). In the first, we masked out the velocity information from the observation vector and only allowed positional information. This modification makes the problem difficult as the agent now needs to estimate these velocities itself. The second modification introduced an additional challenge by adding noise to the positional information communicated to the agent. We sampled the noise from a normal distribution with zero mean and $0.1$ standard deviation. We use GRU as our baseline for this diagnostic task as Morad et al. (2022) reported it to be the best-performing architecture on partially observable classical control tasks, even compared to transformers. We trained for 5M steps, on the two variants of Cartpole,

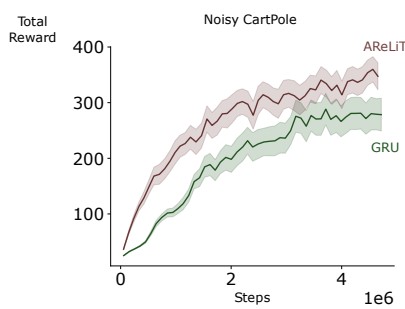

Figure 2: Partially observable CartPole. The vertical axis is the total rewards binned over 10 timesteps and averaged over 27 different seeds $\pm$ standard error. In this experiment, both agents had 1.7M parameters.

two PPO-based agents (Schulman et al., 2017): one using a GRU, and the other using AReLiT. We performed an extensive sweep of the hyperparameters of PPO and the GRU, which is described in Appendix G.2.

Figure 2 summarizes the results from our experiment in Noisy CartPole. The agent based on AReLiT learns faster and finds a better balancing policy than the GRU-based agent. The result on partially observable CartPole (without noise) is qualitatively similar and can be found in Appendix G.2. This result is qualitatively different than the T-Maze because of the different requirements imposed by the environment. In CartPole the agents must integrate information over time to construct a reasonable estimate of the underlying state of the MDP, whereas in T-Maze the agent must learn the cue was important and remember it for a long period of time. These two experiments also demonstrate good performance of AReLiT with two different base RL agorithms: A2C and PPO.

**Mystery Path** In Mystery Path (Pleines et al., 2023), the agent is required to remember multiple cue signals for long periods of time in a 2D pixel-based environments. In this environment, the agent's goal is to reach a target position by traversing through a random invisible path. Episodes have fixed length and the agent is reset back to the start location (along with a feedback observation) upon deviating from the path. We consider two configurations of this environment: MPGrid and the harder MP. In MP, there are six actions and a smoother motion dynamics compared to the easier MPGrid, with grid-like movements and four actions. MPGrid has a maximum episode length of 128, while MP's is 512. Appendix G.3 describes the environment and the configurations considered.

We trained three GTrXL agents with memory sizes $\in \{32, 64, 128\}$ and two AReLiT agents with feature map dimension $\eta \in \{4, 8\}$, and $r = 1$. The architecture sizes for GTrXL and AReLiT were chosen similar to the ones used in the T-Maze experiments. PPO was the base RL agent used. We used a standard agent network architecture (e.g., Mnih et al., 2016; Schulman et al., 2017) for all agents. Details on hyperparameters sweeps can be found in Appendix G.3.

Figure 3 summarizes the main results. Again we report success rate, the percentage of episodes the agent reaches the goal before an episode timeout, calculated over a window of one million steps. Across both configurations (MPGrid and MP) we observe that AReLiT matches the performance of GTrXL-128 when $\eta = 4$ and surpasses GTrXL-128 in mean performance when $\eta = 8$. Also, similar to T-Maze, we observe that reducing the memory size of GTrXL drastically impacts its performance.

We observe again that AReLIT is more computationally efficient than GTrXL. AReLiT-8 uses roughly $55.75$ times fewer operations than GTrXL-128 and it uses $9.84$ times less space. In other words, we observe performance at least as good as GTrXL, in both variants of pixel-based control, at a fraction of the cost.

**Memory Maze** In our final experiment we use a 3D navigation environment called Memory Maze (Pašukonis et al., 2023) that has a fixed horizon and that also requires the agent to remember multiple cue signals for long periods of time. At the beginning of each episode, a new maze is generated randomly and several objects of different colors are distributed across the maze. The agent perceives a $64 \times 64$ RGB image with a colored border indicating the color of the current object of

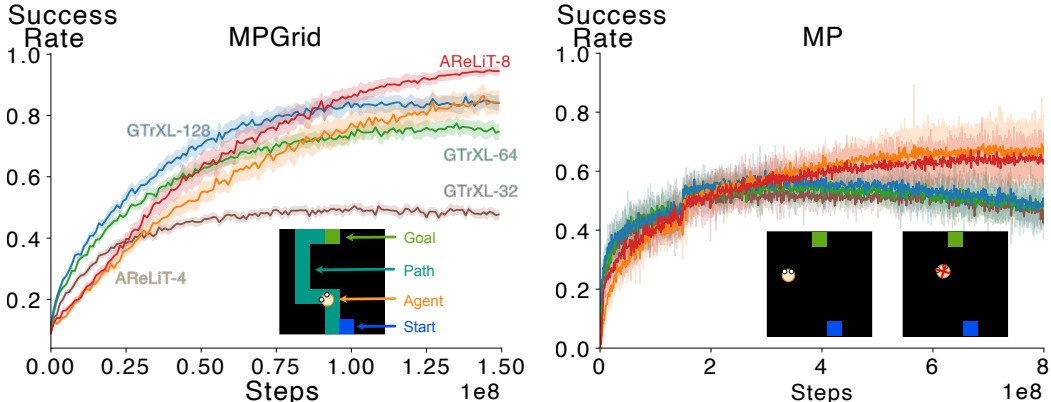

Figure 3: Left: Learning curves in MPGrid (averaged over 15 seeds ± standard error) along with an inset figure showing a possible ground truth maze layout. Right: Learning curves in MP (averaged over 5 seeds ±95% confidence interval) along with inset figure depicting the agent's observation. The agent does not observe the path to goal (left); a red cross is shown as feedback if the agent deviates off from the path, with the agent being reset to the start tile (right).

interest. Once the agent touches the object, it gets a $+1$ reward and the borders' colors changes. The agent's goal is to maximize rewards within the fixed time budget. Thus, the agent must remember the objects' locations to travel through the maze as quickly as possible. Figure 4 provides an illustration of the Memory Maze environment. We report results on the largest maze size, $15 \times 15$, with an episode duration of 4,000 steps. Results for other maze sizes can be found in Appendix H and I.

We trained a GTrXL agent and an AReLiT agent, each with 22M learnable parameters, for 100M steps using the Async-PPO algorithm (Petrenko et al., 2020). The GTrXL agent had a memory size of 256, and the AReLiT agent had a feature map $\eta = 4$ and an approximation hyperparameter $r = 7$. We based our architectures for both the policy and the critic on the work by Petrenko et al. (2020). In this work, a ResNet (He et al., 2016) is used to extract the features from the input image, then a sequence of features are fed into an RNN or a transformer. We detail the hyperparameters used, the architecture sizes, and the tuning strategy in Appendix G.4.

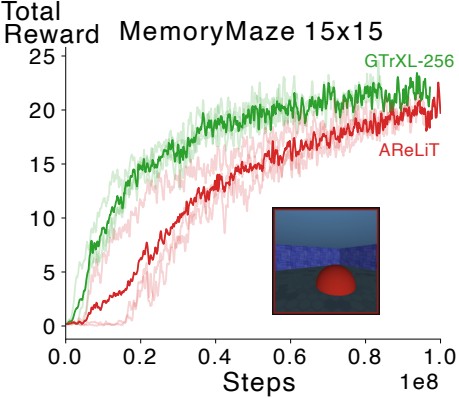

Figure 4: Learning curves of GTrXL-256 and AReLiT in MemoryMaze $15\times15$. The bold lines represent the total episodic reward averaged over an interval of 1M across the three seeds, and the blurred lines represent the individual seeds. The inset plot shows a sample observation.

Figure 4 shows the total episodic reward achieved by our AReLiT-based agent compared with a GTrXL-based agent. The total episodic reward is determined by the number of targets the agent can find within an episode. The asymptotic performance of all the three agents is similar, but the GTrXL-based agent exhibits faster learning early on. Systematic tuning of hyperparameters of our AReLiT-based agent was not feasible due to the significant computational demands of MemoryMaze and the network architectures involved; AReLiT could be improved. This difference could also be an artifact of having few independent runs (three). Regardless, our approach is competitive in large-scale 3D memory/navigation tasks.

Finally, we looked at the agents' utilization of the computational resources. We measured the frames per second (FPS) and the memory usage from 12 AReLiT and GTrXL agents. Overall, AReLiT achieves $535.63 \pm 0.52$ FPS while GTrXL achieves $373.63 \pm 0.49$ FPS, corresponding to a $43.36\%$ improvement. Further, AReLiT uses $52.37\%$ less memory than the GTrXL agent.

## 6 RELATED WORK

Recurrent neural network architectures (Hochreiter & Schmidhuber, 1997; Gao & Glowacka, 2016) are a natural inspiration to our work. They have been applied to a wide range of partially observable RL environments such as Atari 2600 games (Hausknecht & Stone, 2015). However, empirically, RNNs such as LSTMs trained with backpropagation through time often fail to capture long-range dependencies (Khandelwal et al., 2018; Bakker, 2001), which we have also shown in our results.

Gating mechanisms such as the one we used in ReLiT and AReLiT are commonly used in RNNs to control the flow of information and mitigate the impact of vanishing gradients (Hochreiter & Schmidhuber, 1997). Often, scalar gating mechanisms have been applied, such as in the Linear Transformer Peng et al. (2021). However, using a single learned coefficient could be sub-optimal as it controls the flow of past information from each index location in a recurrent state identically. Our results in the T-maze suggest that our gating approach can outperform a single scalar value.

The choice of the feature map $\phi$ can have a significant impact on the overall performance (Schlag et al., 2021). For example, a non-expansive map based on *ELU+1* can be used  Katharopoulos et al. (2020), however, element-wise activation functions are limited in their ability to learn complex non-linear relationships and using them as a feature map limits the memory capacity of the architecture (Schlag et al., 2021). Alternatively, random feature maps can be used to approximate a softmax function  (Peng et al., 2021; Choromanski et al., 2020). Although randomized feature maps are equivalent to softmax function in expectation, they introduce additional variance. Our model is deterministic.

In the context of AReLiT, there are other incremental approaches to approximating large matrices. Incremental Singular Value Decomposition (SVD) (Brand, 2002; 2006) provides a way to perform additive modifications to a low-rank singular value decomposition of a matrix. Previous applications of incremental SVD in RL, however, suggest that sensitivity to the rank parameter is a significant issue (Pan et al., 2017). The rank-one approximation introduced by Ollivier et al. (2015) uses random numbers to approximate a Kronecker delta function producing an unbiased approximation of a matrix represented as a sum of outer products. The use of random numbers, however, introduces variance in the approximation (Cooijmans & Martens, 2019).

Similar to our approach, other methods such as RWKV (Peng et al., 2023), LRU (Orvieto et al., 2023), and S4 (Gu et al., 2021) use recurrent architectures with context-independent inference cost while leveraging parallelization over a sequence. These approaches, however, were only explored within language modeling tasks, with significantly different computation constraints than online RL.

Several works have explore using transformers in RL. Parisotto & Salakhutdinov (2021) used transformers to learn policies in an asynchronous setting relying on policy distillation to make interaction with the environment feasible. Others have explored transformers in model-based, fully-observable RL, such as the TransDreamer architecture which replaces the GRU used inside Dreamer V2 (Hafner et al., 2020) with a transformer (Chen et al., 2022). In the offline RL setting, Chen et al. (2021) re-framed the RL problem as a conditional sequence modeling problem and trained a transformer architecture on a dataset of trajectories (collected from a source RL algorithm).

## 7 CONCLUSION AND FUTURE WORK

Transformers have revolutionized many branches of AI research, but their computational requirements make extension to other domains, such as online RL, difficult. In this paper, we have introduced two recurrent alternatives of the self-attention mechanism is transformers, called Recurrent Linear Transformer (ReLiT) and Approximate Recurrent Linear Transformer (AReLiT). We demonstrate the efficacy of both approaches in a several partially observable reinforcement learning tasks (e.g., T-Maze, MysteryPath, MemoryMaze). The inference cost of our approach is more than 40% cheaper while reducing memory use more than 50%. When compared to a state-of-the-art architecture GTrXL.

Future work could explore algorithmic improvements to AReLiT such as, efficient real-time recurrent learning (Williams & Zipser, 1989) based updates and using different low-rank approximation methods, such as SVD. In addition, previous work has found RNN-based approaches are best in some tasks and transformers better in others. There is much to be understood empirically in partially observable RL.

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

## A    COMPUTATIONAL COMPLEXITIES

The computational complexities of ReLiT, AReLiT, GTrXL and Linear Transformer is presented in Table 1. The computational complexity is for processing a single element in sequence that is presented in a streaming fashion.

Table 1: Space and time complexity of AReLiT, ReLiT, Linear Transformer and GTrXL for a processing a single element in streaming sequence. ($M$: memory size in GTrXL, $d$: representation dimension, $d_k$ feature map dimension in Linear Transformer, $\eta$: feature map hyperparameter in ReLiT and AReLiT, $r$: approximation parameter in AReLiT, $L$: number of encoder layers)

|  | **Space** | **Time** | **Potential Context Length** |
|---|---|---|---|
| GTrXL | $\mathcal{O}(Md)$ | $\mathcal{O}(M\,d^2)$ | $\mathcal{O}(LM)$ |
| Linear Transformer | $\mathcal{O}\left(d_k d\right)$ | $\mathcal{O}\left(d_k d\right)$ | $\infty$ |
| ReLiT | $\mathcal{O}\left(\eta d^2\right)$ | $\mathcal{O}\left(\eta d^2\right)$ | $\infty$ |
| AReLiT | $\mathcal{O}(r\eta d)$ | $\mathcal{O}\left(d^2 + r\eta d\right)$ | $\infty$ |

## B    RECURRENT LINEAR TRANSFORMERS (RELIT)

Algorithm 3 shows the self-attention mechanism introduced in ReLiT. The algorithm introduces a hyper-parameter, $\eta$, and a few learnable parameters, $\mathbf{W}_\beta, \mathbf{W}_\gamma \in \mathbb{R}^{d \times d_h}$ and $\mathbf{W}_{p_1}, \mathbf{W}_{p_2}, \mathbf{W}_{p_3} \in \mathbb{R}^{d \times \eta}$. $\eta$ controls the size of the recurrent states, $\mathbf{C}_t$ and $\mathbf{s}_t$, and also controls the key and the query vectors.

---

**Algorithm 3** Recurrent Linear Transformer (ReLiT) Self-Attention

---

**Input**: $\mathbf{x}_t \in \mathbb{R}^d$, $\mathbf{C}_{t-1} \in \mathbb{R}^{d_h \times \eta d_h}$, $\mathbf{s}_{t-1} \in \mathbb{R}^{\eta d_h}$
**Hyperparameters:** $\eta$
**Parameters**: $\mathbf{W}_K, \mathbf{W}_Q, \mathbf{W}_V, \mathbf{W}_\beta, \mathbf{W}_\gamma \in \mathbb{R}^{d_h \times d}$ and $\mathbf{W}_{p_1}, \mathbf{W}_{p_2}, \mathbf{W}_{p_3} \in \mathbb{R}^{\eta \times d}$

 1: **if** $t = 0$ **then**
 2:     $\mathbf{s}_0 \leftarrow \mathbf{0}, \mathbf{C}_0 \leftarrow \mathbf{0}$.
 3: **end if**

                                                    {Calculate Key}

 4: $\mathbf{k}_t \leftarrow flatten(relu(\mathbf{W}_{p_1}\mathbf{x}_t) \otimes relu(\mathbf{W}_K\mathbf{x}_t))$

                                            {Calculate Query}

 5: $\mathbf{q}_t \leftarrow flatten(relu(\mathbf{W}_{p_2}\mathbf{x}_t) \otimes relu(\mathbf{W}_Q\mathbf{x}_t))$

                                            {Calculate Value}

 6: $\mathbf{v}_t \leftarrow \mathbf{W}_V\mathbf{x}_t$

                                  {Generate Gating Vectors}

 7: $\beta_t \leftarrow \sigma_g(\mathbf{W}_\beta\mathbf{x}_t)$
 8: $\gamma_t \leftarrow flatten(\sigma_g(\mathbf{W}_{p_3}\mathbf{x}_t) \otimes \sigma_g(\mathbf{W}_\gamma\mathbf{x}_t))$

                                        {Update Memory}

 9: $\mathbf{C}_t \leftarrow \left((1-\beta_t) \otimes (1-\gamma_t)\right)\odot\mathbf{C}_{t-1} + \left(\beta_t\odot\mathbf{v}_t\right) \otimes \left(\gamma_t\odot\mathbf{k}_t\right)$
10: $\mathbf{s}_t \leftarrow (1-\gamma_t)\odot\mathbf{s}_{t-1} + \gamma_t\odot\mathbf{k}_t$

                                 {Calculate Attention Vector}

11: $\mathbf{a}_t \leftarrow (\mathbf{C}_t\mathbf{q}_t)/(\mathbf{s}_t\mathbf{q}_t)$

**Output**: $\mathbf{a}_t \in \mathbb{R}^{d_h}$, $\mathbf{C}_t \in \mathbb{R}^{d_h \times \eta d_h}$, $\mathbf{s}_t \in \mathbb{R}^{\eta d_h}$

---

## C  Derivation of AReLiT

In this section, we walk through the derivations to approximate the ReLiT self-attention mechanism. We first start with deriving an approximation for the Kronecker Delta Function and then use these approximation results to derive the AReLit self-attention mechanism.

### C.1  Approximation of Kronecker Delta Function

In this section we derive an approximation of the Kronecker delta function. The Kronecker delta function is defined for integers $m$ and $n$ as:

$$\delta_{mn} = \begin{cases} 1 & \text{if } m = n \\ 0 & \text{if } m \neq n \end{cases}$$

We use a trigonometric identity that is used in computing Fourier series by relating the Kronecker delta function to an integral of a product of two cosine functions (Weisstein). The identity is given by:

$$\delta_{mn} = \frac{1}{\pi} \int_0^{2\pi} \cos(mx) \, \cos(nx) \, dx. \tag{16}$$

We use the Trapezoidal rule to approximate the integral in Equation 16. The trapezoidal rule is a numerical integration method that approximates the integral of a function by dividing the interval into sub-intervals and approximating the function in each sub-interval with a straight line connecting the endpoints. For a function $f(x)$ that is integrable on the interval $[a, b]$, the trapezoidal rule is given by:

$$\int_a^b f(x) \, dx \approx \sum_{k=1}^r \frac{f(x_{k-1}) + f(x_k)}{2} \Delta x, \tag{17}$$

where $\Delta x = \dfrac{b - a}{r}$, $x_k = a + k\Delta x$, and $r$ is the number of sub-intervals used for the integral and it controls the degree of approximation. As $r \to \infty$ the approximation becomes exact. Let $\tilde{\delta}_{mn}$ be the Trapezoidal approximation of the integral defined in Equation 16. We can then write $\tilde{\delta}_{mn}$ as follows:

$$\tilde{\delta}_{mn} = \frac{1}{r} \sum_{i=0}^{r-1} \cos\left(\frac{2\pi i}{r} m\right) \cos\left(\frac{2\pi i}{r} n\right) + \frac{1}{r} \sum_{i=1}^r \cos\left(\frac{2\pi i}{r} m\right) \cos\left(\frac{2\pi i}{r} n\right) \tag{18}$$

Further, in the limit we have: $\lim_{r \to \infty} \tilde{\delta}_{mn} = \delta_{mn}$.

Next, we will simplify the above equation to combine the two summations above into a single one:

$$\tilde{\delta}_{mn} = \frac{1}{r} \sum_{i=0}^{r-1} \cos\left(\frac{2\pi i}{r}m\right) \cos\left(\frac{2\pi i}{r}n\right) + \frac{1}{r} \sum_{i=1}^{r} \cos\left(\frac{2\pi i}{r}m\right) \cos\left(\frac{2\pi i}{r}n\right)$$

Adding and subtracting $\frac{1}{r}(\cos(0)\cos(0) + \cos(2\pi m)\cos(2\pi n))$

$$= \frac{1}{r} \sum_{i=0}^{r-1} \cos\left(\frac{2\pi i}{r}m\right) \cos\left(\frac{2\pi i}{r}n\right) + \cos(2\pi m)\cos(2\pi n)$$

$$+ \frac{1}{r} \sum_{i=1}^{r} \cos\left(\frac{2\pi i}{r}m\right) \cos\left(\frac{2\pi i}{r}n\right) + \cos(0)\cos(0)$$

$$- \frac{1}{r}(\cos(0)\cos(0) + \cos(2\pi m)\cos(2\pi n))$$

$$= \frac{1}{r} \sum_{i=0}^{r-1} \cos\left(\frac{2\pi i}{r}m\right) \cos\left(\frac{2\pi i}{r}n\right) + \cos\left(\frac{2\pi r}{r}m\right) \cos\left(\frac{2\pi r}{r}n\right)$$

$$+ \frac{1}{r} \sum_{i=1}^{r} \cos\left(\frac{2\pi i}{r}m\right) \cos\left(\frac{2\pi i}{r}n\right) + \cos(0)\cos(0)$$

$$- \frac{1}{r}(\cos(0)\cos(0) + \cos(2\pi m)\cos(2\pi n))$$

$$= \frac{2}{r} \sum_{i=0}^{r} \left( \cos\left(\frac{2\pi i}{r}m\right) \cos\left(\frac{2\pi i}{r}n\right) \right) - \frac{1}{r}(\cos(0)\cos(0) + \cos(2\pi m)\cos(2\pi n))$$

Since $m$ and $n$ are integers

$$= \frac{2}{r} \sum_{i=0}^{r} \left( \cos\left(\frac{2\pi i}{r}m\right) \cos\left(\frac{2\pi i}{r}n\right) \right) - \frac{2}{r} \tag{19}$$

We will now present an approximation of the Kronecker delta function that has only the first term in the right hand side of Equation 19. Equation 19 has two terms in the left hand side. The first term is a sum of cosine functions and the second term is a constant. We want approximation of the Kronecker delta function that has only the first term. Let $\hat{\delta}_{mn}$ be an approximation of $\delta_{mn}$ that has only the first term, such that $\hat{\delta}_{mn}$ is defined as follows:

$$\hat{\delta}_{mn} \doteq \frac{2}{r} \sum_{i=0}^{r} \left( \cos\left(\frac{2\pi i}{r}m\right) \cos\left(\frac{2\pi i}{r}n\right) \right) \tag{20}$$

Substituting Equation 20 to Equation 19, we have:

$$\tilde{\delta}_{mn} = \hat{\delta}_{mn} - \frac{2}{r} \tag{21}$$

We can further show that in the limit of $r$, $\hat{\delta}_{mn}$ is equal to $\delta_{mn}$. Applying limit to both sides of the above equation, we have:

$$\lim_{r\to\infty} \tilde{\delta}_{mn} = \lim_{r\to\infty} \hat{\delta}_{mn} - \lim_{r\to\infty} \frac{2}{r} \tag{22}$$

$$= \lim_{r\to\infty} \hat{\delta}_{mn} - 0$$

Since, $\lim_{r\to\infty} \tilde{\delta}_{mn} = \delta_{mn}$, we have:

$$\lim_{r\to\infty} \hat{\delta}_{mn} = \delta_{mn} \tag{23}$$

### C.1.1 USING THE KRONECKER DELTA FUNCTION TO APPROXIMATE RELIT

We start by start by starting ReLiT recurrent state update which we will then approximate using the Kronecker delta approximation introduced above. ReLiT recurrent state update is expressed as follows:

$$\mathbf{C}_t = \big((1 - \beta_t) \otimes (1 - \gamma_t)\big) \odot \mathbf{C}_{t-1} + \big(\beta_t \odot \mathbf{v}_t\big) \otimes \big(\gamma_t \odot \mathbf{k}_t\big) \tag{24}$$

We will now use the approximation of the Kronecker delta function in Equation 20 to approximate the recurrent state update in Equation 24. We start by representing the recurrent state $\mathbf{C}_t$ as a sum of outer products. Starting with Equation 24:

$\mathbf{C}_t = \big((1 - \beta_t) \otimes (1 - \gamma_t)\big) \odot \mathbf{C}_{t-1} + \big(\beta_t \odot \mathbf{v}_t\big) \otimes \big(\gamma_t \odot \mathbf{k}_t\big)$

   Recursively expanding $\mathbf{C}_{t-1}$

$= \Big((\beta_t \odot \mathbf{v}_t) \otimes (\gamma_t \odot \mathbf{k}_t)\Big) + \Big((1 - \beta_t) \otimes (1 - \gamma_t)\Big) \odot \mathbf{C}_{t-1}$

$= \Big((\beta_t \odot \mathbf{v}_t) \otimes (\gamma_t \odot \mathbf{k}_t)\Big)$

$\quad + \Big((1 - \beta_t) \otimes (1 - \gamma_t)\Big) \odot \Big((\beta_{t-1} \odot \mathbf{v}_{t-1}) \otimes (\gamma_{t-1} \odot \mathbf{k}_{t-1}) + \big((1 - \beta_{t-1}) \otimes (1 - \gamma_{t-1})\big) \odot \mathbf{C}_{t-2}\Big)$

$= \Big((\beta_t \odot \mathbf{v}_t) \otimes (\gamma_t \odot \mathbf{k}_t)\Big) + \Big((1 - \beta_t) \otimes (1 - \gamma_t)\Big) \odot \Big((\beta_{t-1} \odot \mathbf{v}_{t-1}) \otimes (\gamma_{t-1} \odot \mathbf{k}_{t-1})\Big)$

$\quad + \Big((1 - \beta_t) \otimes (1 - \gamma_t)\Big) \odot \Big((1 - \beta_{t-1}) \otimes (1 - \gamma_{t-1})\Big) \odot \mathbf{C}_{t-2}$

Since, $(\mathbf{a} \otimes \mathbf{b}) \odot (\mathbf{c} \otimes \mathbf{d}) = (\mathbf{a} \odot \mathbf{c}) \otimes (\mathbf{b} \odot \mathbf{d})$ for arbitrary vectors $\mathbf{a}$, $\mathbf{b}$, $\mathbf{c}$, $\mathbf{d}$, we can rewrite the above equation as follows:

$\mathbf{C}_t = \Big((\beta_t \odot \mathbf{v}_t) \otimes (\gamma_t \odot \mathbf{k}_t)\Big) + \Big(\big((1 - \beta_t) \odot \beta_{t-1} \odot \mathbf{v}_{t-1}\big) \otimes \big((1 - \gamma_t) \odot \gamma_{t-1} \odot \mathbf{k}_{t-1}\big)\Big)$

$\quad + \Big(\big((1 - \beta_t) \odot (1 - \beta_{t-1})\big) \otimes \big((1 - \gamma_t) \odot (1 - \gamma_{t-1})\big)\Big) \odot \mathbf{C}_{t-2}$

   Recursively expanding further

$= \Big((\beta_t \odot \mathbf{v}_t) \otimes (\gamma_t \odot \mathbf{k}_t)\Big) + \Big(\big((1 - \beta_t) \odot \beta_{t-1} \odot \mathbf{v}_{t-1}\big) \otimes \big((1 - \gamma_t) \odot \gamma_{t-1} \odot \mathbf{k}_{t-1}\big)\Big) +$

$+ \Big(\big((1 - \beta_t) \odot (1 - \beta_{t-1}) \odot \beta_{t-1} \odot \mathbf{v}_{t-2}\big) \otimes \big((1 - \gamma_t) \odot (1 - \gamma_{t-1}) \odot \gamma_{t-2} \odot \mathbf{k}_{t-2}\big)\Big) + \dots$

We can introduce variables $\mathbf{l}_i$ and $\mathbf{m}_i$, for $i = 0, 1, \dots, t$ to rewrite the above equation as a sum of outer products:

$$\mathbf{C}_t = \sum_{i=0}^{t} \mathbf{l}_i \otimes \mathbf{m}_i \tag{25}$$

where

$$\mathbf{l}_i = \prod_{j=i+1}^{t} (1 - \beta_j) \odot \beta_i \odot \mathbf{v}_i \tag{26}$$

$$\mathbf{m}_i = \prod_{j=i+1}^{t} (1 - \gamma_j) \odot \gamma_i \odot \mathbf{k}_i \tag{27}$$

Next, we introduce the approximate Kronecker delta function in Equation 20 to approximate the sum of outer products in Equation 25. Continuing from Equation 25, we have:

$$\mathbf{C}_t = \sum_{i=0}^{t} \mathbf{l}_i \otimes \mathbf{m}_i$$

$$= \sum_{j=0}^{t} \sum_{i=0}^{t} \delta_{ij} \mathbf{l}_i \otimes \mathbf{m}_j$$

Replacing $\delta_{i,j}$ with $\hat{\delta}_{i,j}$ we can have an approximation $\tilde{\mathbf{C}}_t$ of $\mathbf{C}_t$ as follows:

$$\mathbf{C}_t \approx \tilde{\mathbf{C}}_t = \sum_{j=0}^{t} \sum_{i=0}^{t} \hat{\delta}_{ij} \mathbf{l}_i \otimes \mathbf{m}_j$$

Using Equation 20

$$= \frac{2}{r} \sum_{j=0}^{t} \sum_{i=0}^{t} \sum_{k=0}^{r} \cos\left(\frac{2\pi k}{r} i\right) \cos\left(\frac{2\pi k}{r} j\right) \mathbf{l}_i \otimes \mathbf{m}_j$$

Rearranging the order of summations

$$= \frac{2}{r} \sum_{k=0}^{r} \sum_{j=0}^{t} \sum_{i=0}^{t} \cos\left(\frac{2\pi k}{r} i\right) \cos\left(\frac{2\pi k}{r} j\right) \mathbf{l}_i \otimes \mathbf{m}_j$$

Let $\omega_k \doteq \cos\left(\frac{2\pi k}{r}\right)$, we then have:

$$\tilde{\mathbf{C}}_t = \frac{2}{r} \sum_{k=0}^{r} \sum_{j=0}^{t} \sum_{i=0}^{t} \cos\left(\omega_k i\right) \cos\left(\omega_k j\right) \mathbf{l}_i \otimes \mathbf{m}_j$$

Since $(ab)(\mathbf{c} \otimes \mathbf{d}) = (a\mathbf{c}) \otimes (b\mathbf{d})$ for scalars $a, b$ and vectors $\mathbf{c}, \mathbf{d}$, we can then write:

$$\tilde{\mathbf{C}}_t = \frac{2}{r} \sum_{k=0}^{r} \sum_{j=0}^{t} \sum_{i=0}^{t} \left(\cos\left(\omega_k i\right) \mathbf{l}_i\right) \otimes \left(\cos\left(\omega_k j\right) \mathbf{m}_j\right)$$

Since $(\mathbf{a} + \mathbf{b}) \otimes \mathbf{c} = \mathbf{a} \otimes \mathbf{c} + \mathbf{b} \otimes \mathbf{c}$ for vectors $\mathbf{a}, \mathbf{b}, \mathbf{c}$, we can then write:

$$\tilde{\mathbf{C}}_t = \frac{2}{r} \sum_{k=0}^{r} \left(\sum_{i=0}^{t} \cos\left(\omega_k i\right) \mathbf{l}_i\right) \otimes \left(\sum_{i=0}^{t} \cos\left(\omega_k i\right) \mathbf{m}_i\right)$$

Using Equation 26 and 27

$$= \frac{2}{r} \sum_{k=0}^{r} \left(\sum_{i=0}^{t} \cos\left(\omega_k i\right) \prod_{j=i+1}^{t} (1 - \beta_j) \odot \beta_i \odot \mathbf{v}_i\right) \otimes \left(\sum_{i=0}^{t} \cos\left(\omega_k i\right) \prod_{j=i+1}^{t} (1 - \gamma_j) \odot \gamma_i \odot \mathbf{k}_i\right) \tag{28}$$

Next, we simplify the above equation and rewrite it in a recurrent form. Let $\tilde{\mathbf{v}}_t^k$ and $\tilde{\mathbf{k}}_t^k$ be defined as:

$$\tilde{\mathbf{v}}_t^k \doteq \sum_{i=0}^{t} \cos\left(\omega_k i\right) \prod_{j=i+1}^{t} (1 - \beta_j) \odot \beta_i \odot \mathbf{v}_i \tag{29}$$

$$\tilde{\mathbf{k}}_t^k \doteq \sum_{i=0}^{t} \cos\left(\omega_k i\right) \prod_{j=i+1}^{t} (1 - \gamma_j) \odot \gamma_i \odot \mathbf{k}_i \tag{30}$$

We can then rewrite Equation 28 in terms of $\tilde{\mathbf{v}}_t^k$ and $\tilde{\mathbf{k}}_t^k$ as follows:

$$\tilde{\mathbf{C}}_t = \frac{2}{r} \sum_{k=0}^{r} \tilde{\mathbf{v}}_t^k \otimes \tilde{\mathbf{k}}_t^k \tag{31}$$

It is possible to regroup the terms in the above equations and derive a recursive relationship of $\tilde{\mathbf{v}}_t^k$ and $\tilde{\mathbf{k}}_t^k$ with respect to $\tilde{\mathbf{v}}_{t-1}^k$ and $\tilde{\mathbf{k}}_{t-1}^k$ as follows:

$$\tilde{\mathbf{v}}_t^k = \sum_{i=0}^{t} \cos(\omega_k i) \prod_{j=i+1}^{t} (1 - \beta_j) \odot \beta_i \odot \mathbf{v}_i$$

$$= \cos(\omega_k t)\beta_t \odot \mathbf{v}_t + \sum_{i=0}^{t-1} \cos(\omega_k i) \prod_{j=i+1}^{t} (1 - \beta_j) \odot \beta_i \odot \mathbf{v}_i$$

Taking common $(1 - \beta_t)$

$$= \cos(\omega_k t)\beta_t \odot \mathbf{v}_t + (1 - \beta_t) \sum_{i=0}^{t-1} \cos(\omega_k i) \prod_{j=i+1}^{t-1} (1 - \beta_j) \odot \beta_i \odot \mathbf{v}_i$$

Replacing with $\tilde{\mathbf{v}}_{t-1}^i$

$$= \cos(\omega_k t)\beta_t \odot \mathbf{v}_t + (1 - \beta_t) \odot \tilde{\mathbf{v}}_{t-1}^k \tag{32}$$

Similarly,

$$\tilde{\mathbf{k}}_t^k = \sum_{i=0}^{t} \cos(\omega_k i) \prod_{j=i+1}^{t} (1 - \gamma_j) \odot \gamma_i \odot \mathbf{k}_i$$

$$= \cos(\omega_k t)\gamma_t \odot \mathbf{k}_t + \sum_{i=0}^{t-1} \cos(\omega_k i) \prod_{j=i+1}^{t} (1 - \gamma_j) \odot \gamma_i \odot \mathbf{k}_i$$

Taking common $(1 - \gamma_t)$

$$= \cos(\omega_k t)\gamma_t \odot \mathbf{k}_t + (1 - \gamma_t) \sum_{i=0}^{t-1} \cos(\omega_k i) \prod_{j=i+1}^{t-1} (1 - \gamma_j) \odot \gamma_i \odot \mathbf{k}_i$$

Replacing with $\tilde{\mathbf{k}}_{t-1}^i$

$$= \cos(\omega_k t)\gamma_t \odot \mathbf{k}_t + (1 - \gamma_t) \odot \tilde{\mathbf{k}}_{t-1}^k \tag{33}$$

Using recursive relationships in Equation 32 and 33, we can now present the final approximation. For a given $r$, we maintain recurrent states $\tilde{\mathbf{v}}_{t-1}^k$ and $\tilde{\mathbf{k}}_{t-1}^k$ for $k = 0, 1, 2, \ldots, r$. For $\omega_k \doteq \frac{2\pi k}{r}$, and assuming $\tilde{\mathbf{v}}_0^i$ and $\tilde{\mathbf{k}}_0^i$ are initialized as zeros, the recurrent updates to $\tilde{\mathbf{v}}_t^i$ and $\tilde{\mathbf{k}}_t^i$ and further the approximation to $\mathbf{C}_t$ are given by:

$$\mathbf{C}_t \approx \tilde{\mathbf{C}}_t = \frac{2}{r} \sum_{k=0}^{r} \tilde{\mathbf{v}}_t^k \otimes \tilde{\mathbf{k}}_t^k \tag{34}$$

where, for $k = 0, 1, 2, \ldots, r$ we have:

$$\tilde{\mathbf{v}}_t^k \doteq \cos(\omega_k t)\beta_t \odot \mathbf{v}_t + (1 - \beta_t) \odot \tilde{\mathbf{v}}_{t-1}^k \tag{35}$$

$$\tilde{\mathbf{k}}_t^k \doteq \cos(\omega_k t)\gamma_t \odot \mathbf{k}_t + (1 - \gamma_t) \odot \tilde{\mathbf{k}}_{t-1}^k \tag{36}$$

Since $\lim_{r \to \infty} \hat{\delta}_{mn} = \delta_{mn}$, it follows that $\lim_{r \to \infty} \tilde{\mathbf{C}}_t = \mathbf{C}_t$. Unlike Equation 24, Equation 35 and 36 define a recurrence over vectors instead of matrices, and if $r << d$, the recurrence is much more efficient in space than the recurrence in Equation 24. We leave it to future work to formally derive the approximation error. In Section E we show the approximation error with a synthetic error under different values of $r$.

Lastly, since the current state $\tilde{\mathbf{C}}_t$ could be represented as a sum of outer products in a non-recurrent manner, we can avoid explicitly calculating $\tilde{\mathbf{C}}_t$ and instead calculate the attention output $\mathbf{a}_t$ as follows:

$$\mathbf{a}_t \doteq \frac{\sum_{k=0}^{r} \tilde{\mathbf{v}}_t^k \left( \left( \tilde{\mathbf{k}}_t^k \right)^{\mathsf{T}} \mathbf{q}_t \right)}{2r(\mathbf{s}_t^{\mathsf{T}} \mathbf{q}_t)} \tag{37}$$

# D  APPROXIMATE RECURRENT LINEAR TRANSFORMER (AReLiT)

Algorithm 4 shows the Approximate Recurrent Linear Transformer (AReLiT). We highlight changes from Algorithm 3 in blue. The algorithm maintains a set of vectors $\tilde{\mathbf{k}}_{t-1}^0, ..., \tilde{\mathbf{k}}_{t-1}^r \in \mathbb{R}^{\eta d_h}$, $\tilde{\mathbf{v}}_{t-1}^0, ..., \tilde{\mathbf{v}}_{t-1}^r \in \mathbb{R}^{d_h}$, and $\mathbf{s}_{t-1} \in \mathbb{R}^{\eta d_h}$ as the recurrent state at a given time-step $t$. The number of vectors stored could be controlled by modifying the hyperparameter $r$, which should ideally be set to a small value. The key, query, and value vectors are calculated similarly to ReLiT. The recurrent state update is modified to use the approximation in Equation 34. At each time step, the recurrent vectors are updated using element-wise vector multiplication and addition operations (lines 10-14). The operation on each recurrent vector could be executed in parallel. The attention output is calculated without ever explicitly calculating $\tilde{\mathbf{C}}_t$ (lines 16-18).

---

**Algorithm 4** Approximate Recurrent Linear Transformer (AReLiT) Self-Attention (Streaming Data)

---

**Input**: $\mathbf{x}_t \in \mathbb{R}^d$, $\tilde{\mathbf{k}}_{t-1}^0, ..., \tilde{\mathbf{k}}_{t-1}^r \in \mathbb{R}^{\eta d_h}$, $\tilde{\mathbf{v}}_{t-1}^0, ..., \tilde{\mathbf{v}}_{t-1}^r \in \mathbb{R}^{d_h}$, and $\mathbf{s}_{t-1} \in \mathbb{R}^{\eta d_h}$
**Hyperparameters**: $\eta$ and $r$.
**Parameters**: $\mathbf{W}_K, \mathbf{W}_Q, \mathbf{W}_V, \mathbf{W}_\beta, \mathbf{W}_\gamma \in \mathbb{R}^{d_h \times d}$ and $\mathbf{W}_{p_1}, \mathbf{W}_{p_2}, \mathbf{W}_{p_3} \in \mathbb{R}^{\eta \times d}$

1: Assume $\mathbf{s}_0 \leftarrow 0, \mathbf{C}_0 \leftarrow 0$.

         {Calculate Key}
2: $\mathbf{k}_t \leftarrow \textit{flatten}(\textit{relu}(\mathbf{W}_{p_1}\mathbf{x}_t) \otimes \textit{relu}(\mathbf{W}_K\mathbf{x}_t))$

         {Calculate Query}
3: $\mathbf{q}_t \leftarrow \textit{flatten}(\textit{relu}(\mathbf{W}_{p_2}\mathbf{x}_t) \otimes \textit{relu}(\mathbf{W}_Q\mathbf{x}_t))$

         {Calculate Value}
4: $\mathbf{v}_t \leftarrow \mathbf{W}_V\mathbf{x}_t$

         {Generate Gating Vectors}
5: $\beta_t \leftarrow \sigma_g(\mathbf{W}_\beta\mathbf{x}_t)$
6: $\gamma_t \leftarrow \textit{flatten}(\sigma_g(\mathbf{W}_{p_3}\mathbf{x}_t) \otimes \sigma_g(\mathbf{W}_\gamma\mathbf{x}_t))$

         {Update Memory}
7: **for** $i \leftarrow 0$ to $r$ **in parallel, do**
8:     $\omega_i \leftarrow (2\pi i)/r$
9:     $\tilde{\mathbf{v}}_t^i \leftarrow \tilde{\mathbf{v}}_{t-1}^i \odot (1 - \beta_t) + \cos(\omega_i t)(\beta_t \odot \mathbf{v}_t)$
10:    $\tilde{\mathbf{k}}_t^i \leftarrow \tilde{\mathbf{k}}_{t-1}^i \odot (1 - \gamma_t) + \cos(\omega_i t)(\gamma_t \odot \mathbf{k}_t)$
11: **end for**
12: $\mathbf{s}_t \leftarrow (1 - \gamma_t) \odot \mathbf{s}_{t-1} + \gamma_t \odot \mathbf{k}_t$

         {Calculate Attention Vector}
13: $\mathbf{a} \leftarrow \sum_{i=0}^r \tilde{\mathbf{v}}_t^i \left( \tilde{\mathbf{k}}_t^{i\mathsf{T}}\mathbf{q}_t \right)$
14: $\mathbf{b} \leftarrow 2r(\mathbf{s}_t^\mathsf{T}\mathbf{q}_t)$
15: $\mathbf{a}_t \leftarrow \mathbf{a}/\mathbf{b}$

**Output**: $\mathbf{a}_t \in \mathbb{R}^{d_h}$, $\tilde{\mathbf{k}}_t^0, ..., \tilde{\mathbf{k}}_t^r \in \mathbb{R}^{\eta d_h}$, $\tilde{\mathbf{v}}_t^0, ..., \tilde{\mathbf{v}}_t^r \in \mathbb{R}^{d_h}$, and $\mathbf{s}_t \in \mathbb{R}^{\eta d_h}$

---

# E  EFFECT OF $r$ ON THE QUALITY OF APPROXIMATION IN AReLiT

We empirically evaluate the effect of $r$ on the quality of the approximation of the current state matrix $\mathbf{C}_t$. Ideally, we want to set $r$ to a small value as the space complexity of AReLiT is directly proportional to $r$. We consider a synthetic example where the value $\mathbf{v}_t$ and key $\mathbf{k}_t$ at each time step are sampled randomly from a normal distribution. We set the embedding dimension $d$ to 128 and randomly sample values and keys for 100 timesteps. Instead of using vectors $\gamma_t$ and $\beta_t$ for gating at every timestep, we use a constant value $c$. We then compare the difference between the current state matrix $\mathbf{C}_t$ computed using the exact method in Equation 5, with the current state matrix $\tilde{\mathbf{C}}_t$ computed using the approximate method in Equation 34 at the 100th time-step. We use the Frobenius norm to measure the difference between the two matrices. We repeat the experiment for different

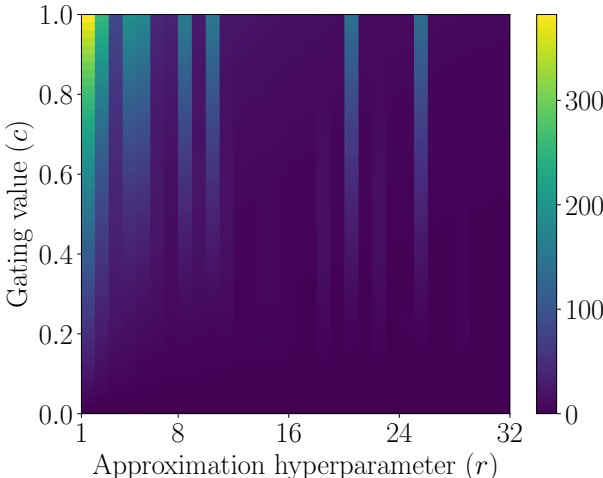

Figure 5: Error in approximating the current state $\mathbf{C}_t$ for different values $r$ and gating at $t = 100$ for randomly sampled values and keys.

values of $r$ and $c$. For each configuration, we report the mean error across 50 independent runs. Figure 5 shows the results of this experiment. We observe that the error in approximation decreases with increasing value of $r$. For most values of $r$ and $c$, the approximation error is low. This is useful since it allows us to set $r$ to a small value, thereby reducing the space complexity of the model. In fact, in the largest experiments described in this thesis, we set $r$ to 7. Interestingly, we observe periodic bands in the error plot. It is possible that this is due to the periodicity of the cosine functions used in the attention mechanism. We leave further exploration around the theoretical nature of the error in approximation for future work.

## F  PARALLELIZATION OVER AN INPUT SEQUENCE

Transformers are naturally designed for parallelism over a sequence of input data, as the self-attention operation does not have dependencies between different parts of the input sequence. It is essential to consider the parallelizability of transformer architectures, when the input sequence is presented in a batched fashion. Such a scenario is common in practice, as most existing actor-critic approaches such as PPO and A2C (Schulman et al., 2017; Mnih et al., 2016) estimate gradient updates to the actor and critic using batches of trajectories collected through agent-environment interactions. Furthermore, most modern hardware accelerators, such as GPUs and TPUs, excel in handling parallelizable algorithms, and parallelization is vital for effectively training large models.

Extension of Algorithm 3 and 4 to accommodate parallelization over a sequence of inputs is straightforward, depending on whether the computation has dependencies on the previous state or not. The majority of the computations in both algorithms, which involve calculating keys, queries, values, gating vectors, and the attention vector, do not depend on the previous state and can be parallelized over the sequence. The only part of the algorithm that depends on the previous state is the update of the current state. In Algorithm 3, this is done from lines 13-14, and in Algorithm 4, from lines 10-15. The update of the current state in both algorithms is implemented as a first order recurrence. This operation is parallelizable as such recurrences could be expressed as an associtiave binary operations (see Blelloch, 1990). In our implementation, we used the *associative_scan* operation in Jax to parallelize ReLiT and AReLiT over an input sequence.

# G  ADDITIONAL EXPERIMENT DETAILS

## G.1  T-MAZE

**Environment Description:** The T-Maze environment considered in this paper is similar to the one proposed by Bakker (2001). Figure 6 shows two possible episodes in the T-Maze environment. At each timestep, the agent receives a 16-bit binary observation. The first two bits correspond to the cue signal which is either 01 or 10 at the first timestep of an episode, depending on whether the reward is located at the left or right turn at the intersection, respectively. The cue bits are zero in all other timesteps. We consider the largest possible corridor length as 200. To encode the corridor information, the agent additionally receives 8-bit gray code encoding of its current location. The gray code encoding is zero at the beginning of an episode and is updated at each timestep. To make the problem more challenging, we added 6 noisy distractor bits to the observation. The distractor bits are sampled uniformly at random at each timestep. The agent can take one of the four possible discrete actions at each timestep: up, down, left, or right. The agent receives a reward of -0.1 at each non-terminal timestep. At termination, the agent receives a reward of +4 for taking the correct turn and a reward of -1 for taking an incorrect turn. The reward of +4 is chosen to encourage the agent to take the correct turn at the intersection. The difficulty of this environment can be increased by increasing the corridor length. Increasing the corridor length requires the agent to remember the signal for a longer number of timesteps. Since the agent's observations include distractor bits, the agent also needs to learn to ignore the distractor bits and focus on the cue signal.

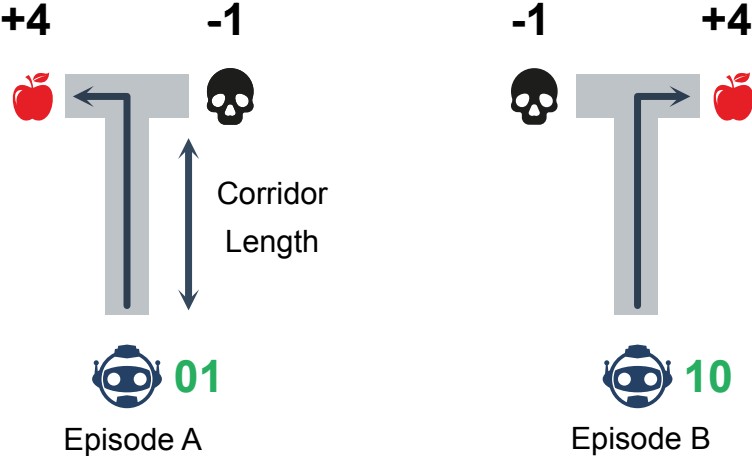

Figure 6: The T-Maze environment. The agent has to remember a binary cue (denoted by green text), shown only at the beginning of the episode, in order to take the correct turn at the intersection and receive a positive reward. The figure shows two possible episodes and the optimal path an agent must take. The agent's current location is provided as gray code encoding in the observation, along with distractor signals. The corridor length could be varied to increase the difficulty of the problem.

**Hyperparameters and Tuning Strategy:** We include the architecture configuration for each of the 5 architectures in Table 3. Our hyperparameter tuning strategy is as follows: We train 5 seeds per architecture for each corridor length in 120-200 and hyperparameter configuration for 5M steps. We identify the best hyperparameter configuration according to the best mean success rate in the last 100K steps across all corridor lengths.

A few additional details are worth reporting for the purposes of reproducibility. We conducted all experiments using Python and implemented the agents using the Jax library (Bradbury et al. (2018)). We used the GTrXL implementation from the DIEngine library (engine Contributors, 2021). Each agent is trained using 16-core machine with 12GB RAM. The network weights are initialized using orthogonal initialization (Saxe et al. (2013)). A single run using the slowest architecture takes around 20 hours to complete.

Table 2: Hyperparameters and sweeps for the T-Maze experiments.

| Hyperparameter | Value |
|---|---|
| Learning Rate | [0.001, 0.0001 0.0005, 0.00001, 0.00005] |
| Discount Factor ($\gamma$) | 0.99 |
| Advantage Estimation Coefficient ($\lambda$) | 0.95 |
| Entropy Coefficient | [0.1, 0.01, 0.001, 0.0001, 0.00001] |
| Value Loss Coefficient | 0.5 |
| Rollout Len | 256 |
| Num of Envs | 8 |
| Batch Size (Rollout Len $\times$ Num of Envs) | 2048 |
| Actor Layer Dimension | 128 |
| Critic Layer Dimension | 128 |

Table 3: Architecture configuration for LSTM, GRU, GTrXL, ReLiT and AReLiT for T-Maze and MysteryPath experiments.

| Hyperparameter | LSTM | GRU | GTrXL | ReLiT | AReLiT |
|---|---|---|---|---|---|
| Embedding Dimension ($d$) | 600 | 680 | 128 | 128 | 128 |
| Hidden Dimension | 1200 | 1360 | N/A | N/A | N/A |
| Num Heads | N/A | N/A | 4 | 4 | 4 |
| Head Dim ($d_h$) | N/A | N/A | 64 | 64 | 64 |
| Num Layers ($L$) | 1 | 1 | 4 | 4 | 4 |
| Memory Size ($M$) | N/A | N/A | [128, 256] | N/A | N/A |
| Projection Hyperparameter ($\eta$) | N/A | N/A | N/A | 4 | [4,8] |
| Approximation Hyperparameter ($r$) | N/A | N/A | N/A | N/A | 1 |
| Actor Layer Dimension | 128 | - | - | - | - |
| Critic Layer Dimension | 128 | - | - | - | - |

## G.2  PARTIALLY OBSERVABLE CARTPOLE

Table 4 shows PPO hyperparameters used for CartPole experiments. We show additional results for the partially observable CartPole environment when no noise is added to the observation vector in figure 7

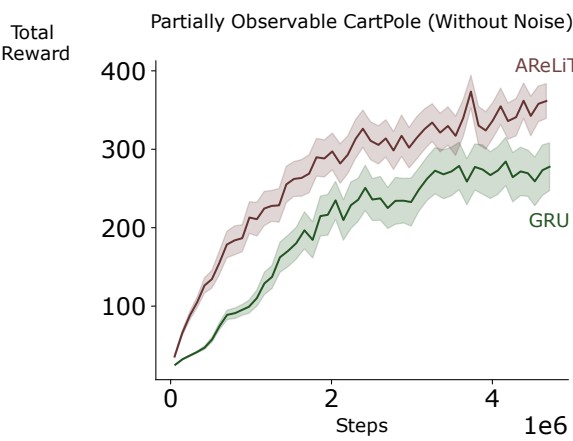

Figure 7: Non-noisy Partially Observable CartPole

Table 4: Hyperparameters and sweeps for the CartPole experiments.

| Hyperparameter | Value |
|---|---|
| Learning Rate | [0.01, 0.001, 0.0001, 0.00001] |
| Discount Factor ($\gamma$) | 0.99 |
| Advantage Estimation Coefficient ($\lambda$) | 0.9 |
| Entropy Coefficient | 0.0 |
| Value Loss Coefficient | 1.0 |
| Rollout Len | 1024 |
| Num of Envs | 1 |
| Batch Size (Rollout Len $\times$ Num of Envs) | 1024 |
| Number of Epochs | 10 |
| PPO Clip Ration | 0.2 |
| Max Gradient Norm | 0.5 |

### G.3 MYSTERY PATH

**Environment Description:** Pleines et al. ( 2023) introduced the Mystery Path environment as part of the Memory Gym benchmark, which aimed to test agents' abilities to memorize many events over an episode. The Mystery Path is a $7 \times 7$ grid environment with pixel-based observations. At the beginning of each episode, the start position of the agent, the origin, is sampled from the grid's borders. Then, the target position is sampled from the grid's borders on the opposite side of the origin. A randomly generated path then connects both the origin and the goal. Figure 8a shows an example of a generated origin, goal, and path. The agent's observation, shown in Figure 8b, is a $64 \times 64$ RGB image containing the origin, the target, and the agent. The agent gets a $+1$ reward when it reaches the goal and a $0.1$ reward when visiting a new tile on the path to the goal. If the agent falls off the path, as in Figure 8c, a red cross appears as visual feedback, and the agent returns to the origin. The reward is zero in all other timesteps. We consider two variants of this environment, MPGrid and MP. MPGrid has maximum episode length of 128, uses grid-like movements and 4 possible actions (left, right, up and down). On the other hand, MP has a maximum episode length of 512, has smoother movements, and a larger action space that allows diagonal movements.

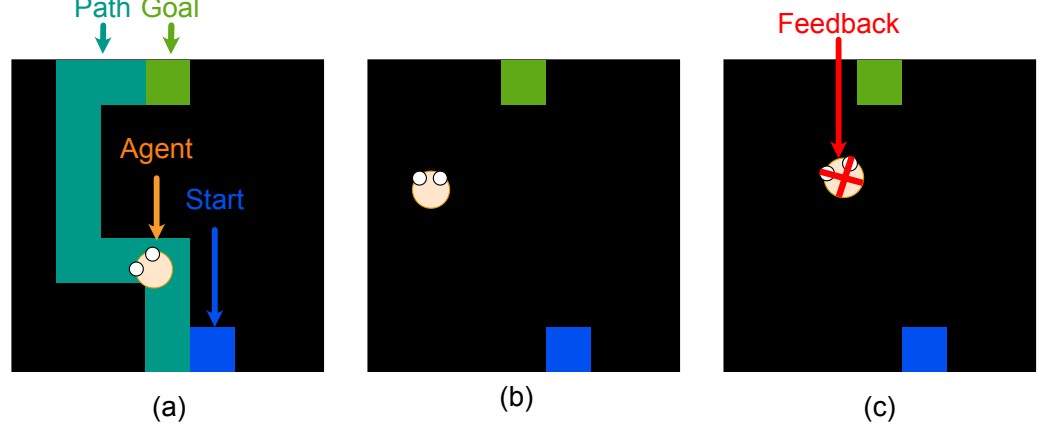

Figure 8: A visualization of the Mystery Path environment.

**Hyperparameters and Tuning Strategy:** The architecture sizes used for Mystery Path experiments are kept same as in Table 3, however, we used actor and critic layer dimension of 256. We detail the hyperparameters used for the PPO algorithm that used for training the agents in the Mystery Path environment in Table 5. We tune learning rate and entropy coefficient for the sweeps mentioned in Table 5. Our hyperparameter tuning strategy is as follows: we train 3 seeds per architecture for each the hyperparameter configuration for 60M steps in the Mystery Path Grid environment. Finally, we

Table 5: Hyperparameters and sweeps for Mystery Path experiments.

| Hyperparameter | Value |
|---|---|
| Learning Rate | [0.0025, 0.00025, 0.000025] |
| Discount Factor ($\gamma$) | 0.99 |
| Advantage Estimation Coefficient ($\lambda$) | 0.95 |
| Entropy Coefficient | [0.03, 0.003, 0.0003, 0.00003] |
| Number of Epochs | 3 |
| Rollout Length | 128 |
| Sequence Length | 128 |
| Number of Env | 128 |
| Batch Size (Sequence Length $\times$ Number of Env) | 16384 |
| Number of Mini Batches | 8 |
| Number of Epochs | 3 |
| PPO Clip Ratio | 0.2 |
| Max Gradient Norm | 4 |
| Value Function Coefficient | 0.5 |

identify the best hyperparameter configuration according to the best episodic reward in the last 1M training steps.

### G.4  MEMORY MAZE

**Environment Description:** The Memory Maze environment evaluates an agent's long-term memory capabilities in a partially observable RL setting. Figure 9 illustrates this environment. The agent's observation at each time-step is an image with $64 \times 64$ RGB pixels, and the action space is discrete. In each episode, the agent starts in a randomly generated maze containing several objects of different colors. The agent's objective is to find the target object of a specific color, indicated by the border color in the observation image. Upon successfully touching the correct object, the agent receives a +1 reward, and the next random object is chosen as the new target. If the agent touches an object of the wrong color, there is no effect on the environment. The maze layout and object locations remain constant throughout the episode. Each episode lasts for a fixed amount of time. Since the maze layout is randomized at each episode, the agent must learn to quickly remember the maze layout, the target object locations, and the paths leading to them.

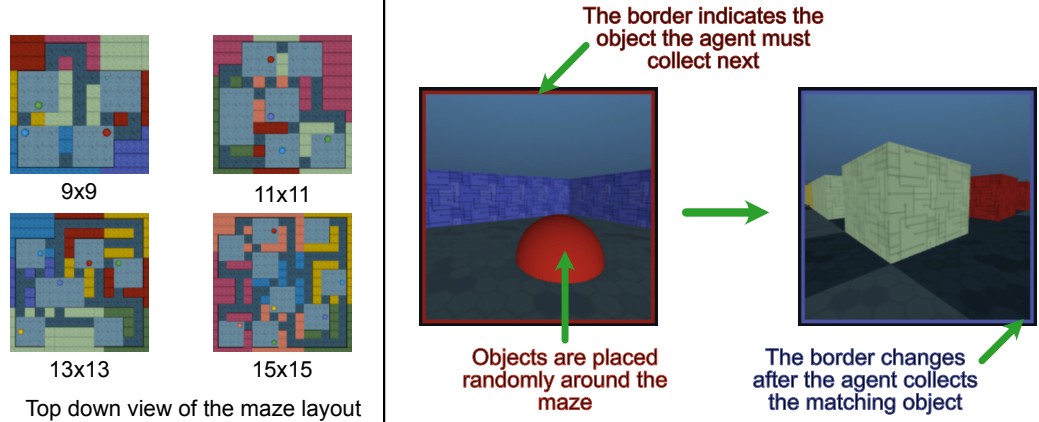

Figure 9: The Memory Maze environment. On the left, we show a possible maze layout for all four Memory Maze configurations. The maze layout is randomized at each episode. On the right, we show two sample observations that the agent receives. The agent's observation at each time-step is $64 \times 64$ RGB pixels and the action space is discrete. The border color of the observation image indicates the target object color which the agent needs to find to receive a reward. After collecting the object, the border color changes, indicating the next target object. The episode lengths are fixed depending on the Memory Maze configuration, with larger configurations having longer episodes.

**Hyperparameters and Tuning Strategy:** We include the details of the Memory Maze experiments. All of the experiments in that section were implemented using asynchronous PPO implementation from Sample Factory library (Petrenko et al. (2020)). We started with the default hyperparameters for the DMLab lab experiments in Schulman et al. (2015), and finetuned the learning rate and entropy coefficient. For each of LSTM, GTrXL and AReLiT, to tune the learning rate and entropy coefficient, we run a sweep for three seeds for 15M steps in the Memory Maze $11 \times 11$ environment. We average the results for the last 1M steps across the three seeds and select the best hyperparameter according to total episodic reward. Using the best-identified hyperparameter, we generate the final results for 100M steps for each of the three seeds. We detail the hyperparameters along with the sweeps for the learning rate and entropy coefficient in Table 6. We include the architecture configuration for each of the 3 architectures in Table 7.

Table 6: Hyperparameters and sweeps for Memory Maze experiments.

| Hyperparameter | Value |
| --- | --- |
| Learning Rate | [0.0025, 0.00025, 0.000025] |
| Discount Factor ($\gamma$) | 0.99 |
| Advantage Estimation Coefficient ($\lambda$) | 0.95 |
| Entropy Coefficient | [0.03, 0.003, 0.0003] |
| Number of Epochs | 1 |
| Rollout Length | 200 |
| Sequence Length | 100 |
| Batch Size | 3200 |
| PPO Clip Ratio | 0.1 |
| PPO Clip Value | 1 |
| Max Gradient Norm | 4 |
| Value Function Coefficient | 0.5 |
| Number of Workers | 32 |
| Number of Envs per Worker | 2 |

Table 7: Architecture configuration for GTrXL and AReLiT for Memory Maze experiments.

| Hyperparameter | GTrXL | AReLiT |
| --- | --- | --- |
| Embedding Dimension ($d$) | 512 | 512 |
| Num Heads | 8 | 8 |
| Head Dim ($d_h$) | 64 | 64 |
| Num Layers ($L$) | 4 | 4 |
| Memory Size ($M$) | 256 | N/A |
| Projection Hyperparameter ($\eta$) | N/A | 4 |
| Approximation Hyperparameter ($r$) | N/A | 7 |

## H    ADDITIONAL LEARNING CURVES ON SMALLER MEMORY MAZE CONFIGURATIONS

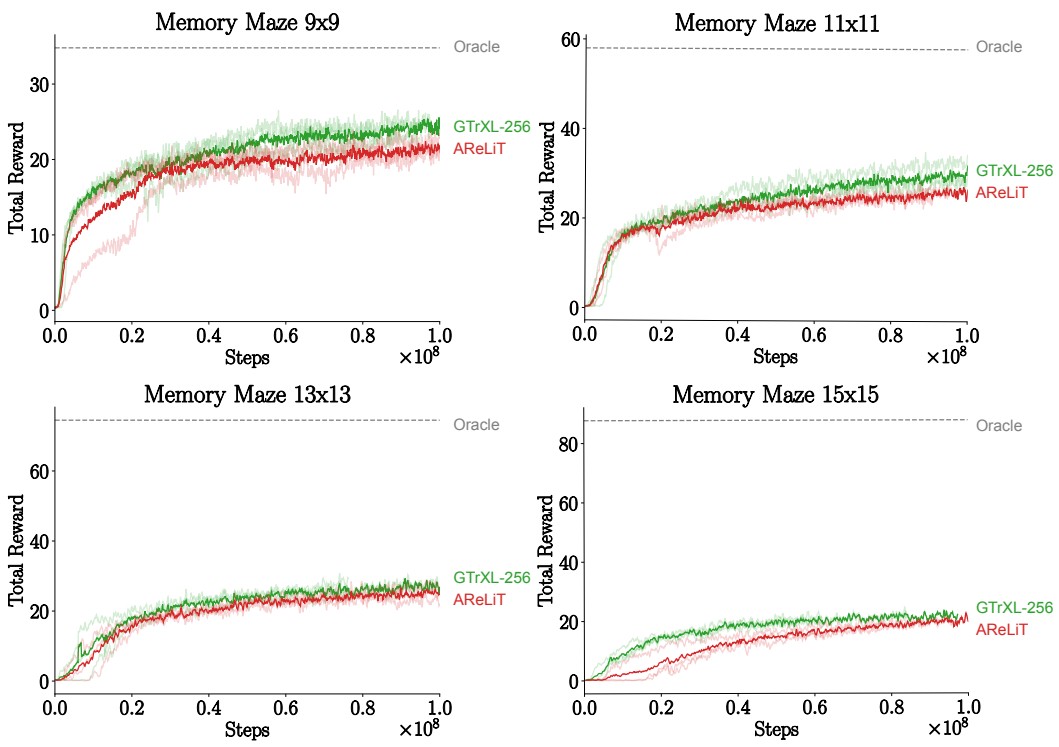

Figure 10: Learning curves of GTrXL and AReLiT agents in the Memory Maze environment. The x-axis represents the number of environment steps, and the y-axis represents the total reward in an episode. Each agent is trained with 3 different random seeds. The bold lines represent the mean return across the 3 seeds, and the blurred lines represent the individual seeds. Each point is the average episodic reward over 1M environment steps. The dotted grey line represents the performance of an oracle agent that has access to the entire maze layout, target object locations and paths leading to them.

## I    EVALUATING IMPACT OF GTRXL'S CONTEXT IN MEMORY MAZE

This experiment evaluates the impact of GTrXL's context length in the Memory Maze environment. We showed earlier that GTrXL's performance is bottlenecked by the memory size in T-Maze. Our hypothesis is that a similar conclusion should hold in the Memory Maze environment. We expect that GTrXL with a larger memory size would outperform GTrXL with a smaller memory size. We should also be able to show that an AReLiT would outperform a GTrXL with a small memory size. To investigate this, we train two additional GTrXL agents with memory sizes of 64 and 128 in the Memory Maze $13 \times 13$ environment.

The learning curves of training the three memory sizes of GTrXL and AReLiT in the Memory Maze $13 \times 13$ environment is shown in Figure 11. Asymptotically, all four agents achieve similar performance. The individual learning curves, however, indicate that the GTrXL-64 agent is slower to converge than the GTrXL-128 and GTrXL-256 agents.

The results failed to provide sufficient evidence to support our hypothesis. The performance obtained by the three agents does not appear to be different. This observation leads us to the following speculation: the Memory Maze environment is too difficult for the agents to be able to utilize their long-term memory capabilities. The reward signal is sparse, which might make it difficult for the agent to learn long-term dependencies. It is also possible that learning long-term dependencies in

navigation tasks is harder, in general, and longer training is necessary for the benefits of long-term memory to show.

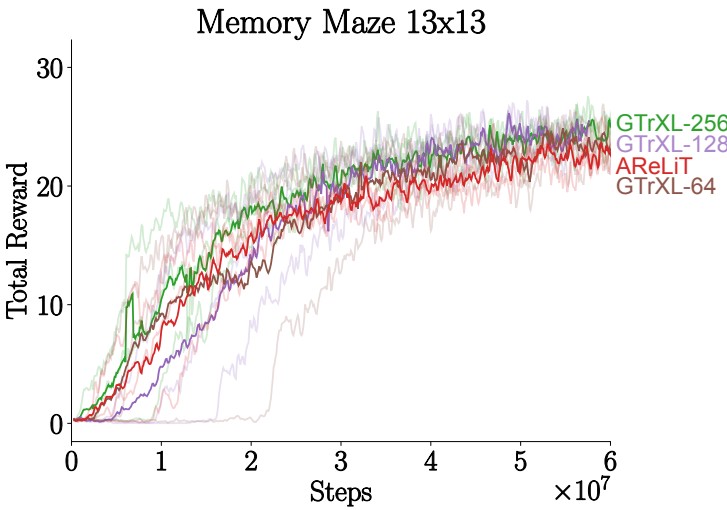

Figure 11: Learning curves of GTrXL agents with different memory sizes in the Memory Maze $13 \times 13$ environment. The x-axis represents the number of environment steps, and the y-axis represents the total reward in an episode. Each agent is trained with 3 different random seeds. The bold lines represent the mean return across the 3 seeds, and the blurred lines represent the individual seeds. Each point is the average episodic reward over 1M environment steps.

## J    Latency Measurements

In this section we provide additional empirical evidence of the computational efficiency of our proposed approach, by comparing the latency of forward pass using GTrXL and AReLiT. We measure the time required in milliseconds (ms) to do a forward pass in two scenarios: (1) processing single element in streaming sequence, (2) processing an entire sequence in parallel. We configure the architecture sizes of GTrXL and AReLiT according to the values used by Parisotto et al. (2020): 12 layers, 8 heads, $d_h = 64$, $d = 256$. We collected all data in a single Google Cloud instance with NVIDIA A100 GPU, 12 CPUs and 80GB RAM.

First, we compare the time required in milliseconds (ms) to do a forward pass using a single element in streaming sequence. We present the results of these comparisons in Figure 12a. According to Dai et al. (2019), XL attention used in the GTrXL architecture has a limited context. The context length of XL attention, how far back in time the transformer architecture can remember, is $\mathcal{O}(ML)$, where $L$ is the number of layers and $M$ is the memory size. We measure the impact of increasing the context length (varying $M$) of GTrXL (x-axis) on the latency to do a single forward pass (y-axis). AReLiT does not explicit hyper-parameter that allows controlling the context length, and the use of a recurrent hidden state allows for a potentially unlimited context. Therefore, we consider three AReLiT architectures with feature map hyper-parameter $\eta \in [4, 8, 16]$, and plot it as a straight line. We observe that the gap between GTrXL and AReLiT increases dramatically with increasing context length.

Next, we measure the time required to do a forward pass over a batch, that is process an entire input sequence in parallel. We present the results of these comparisons in Figure 12b. We vary the length of the input sequence (x-axis) and measure the time required to do a forward pass over the entire sequence (y-axis). We consider two GTrXL architectures with memory size $M \in [128, 512]$. We consider three AReLiT architectures with $\eta \in [4, 8, 16]$. We observe that the gap between GTrXL and AReLiT increases dramatically with increasing sequence length.

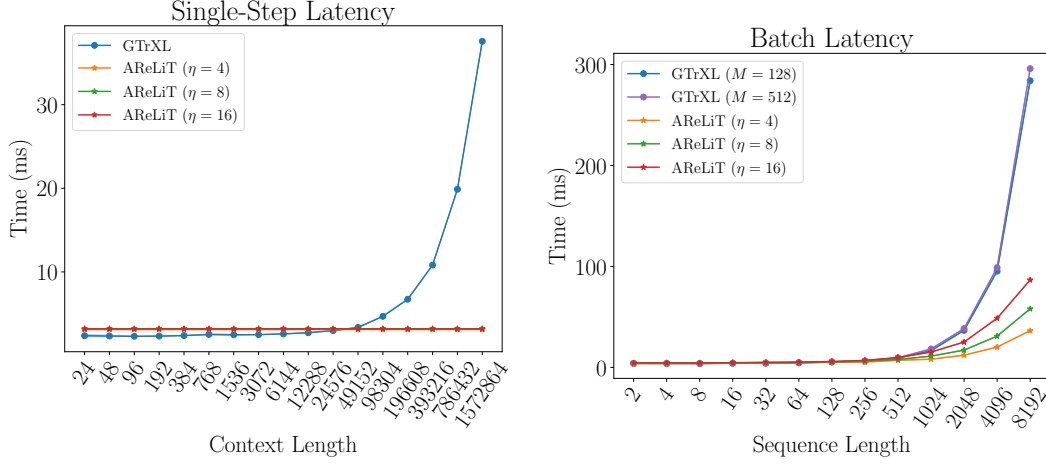

(a) Time (ms) for processing a single element in sequence.

(b) Time (ms) for processing an entire sequence in parallel.

Figure 12: Latency measurements for GTrXL and AReLiT. Each point is averaged over 100 independent runs, and the shaded region is the standard error.

## K ABLATION STUDY

In this section we present ablations for each for each of the three modifications proposed to the Linear Transformer architecture in AReLiT. We present these ablations in Figure 13. We conducted each ablation on the T-Maze environment with corridor length set to 200. We use the same hyperparameter tuning strategy as described in Section G.1, but select the best hyperparameter only on corridor length 200. We report the success rate while training an agent for 5M steps, over 50 seeds.

Our first ablation evaluates the impact of different gating mechanisms in AReLiT. We compare AReLiT's gating mechanism with scalar gating mechanism proposed by Peng et al. (2021) in Figure 13a. Peng et al. (2021) introduced a scalar gating mechanism to the Linear Transformer architecture which allows the Linear Transformer to be trained on long sequences in state-full fashion. We observe that our proposed gating mechanism outperforms the scalar gating mechanism.

The second ablation evaluates the impact of different feature map $\phi$ in AReLiT. We consider two alternatives, proposed in the existing literature. The first uses an element-wise feature map $ELU + 1$ (Clevert et al., 2016), which was used originally in the Linear Transformer architecture. The second is the deterministic parameter free projection (DPFP) introduced by Schlag et al. (2021), which was shown to outperform exisiting feature map approaches in language modelling tasks. We present these results in Figure 13b. We observed that our proposed feature map outperform both of these methods.

The third ablation compares AReLiT's approximation to an alternative incremental low-rank approximation method. We consider the rank-1 trick introduced by Ollivier et al. (2015). The rank-1 tricks approximates a Kronecker delta function using random signs drawn from a uniform distribution. Similar to our proposed approximation approach, the rank-1 trick could be applied to derive incremental updates to a low-rank decomposition of a matrix. We derived an approximation using the rank-1 trick and compared it to our proposed approximation (with $r = 1$) in Figure 5. We observe that our proposed approximation approach outperforms the rank-1 trick.

Additionally, we compare AReLiT with Linear Transformer in Figure 13d. We consider the state-full Linear Transformer approach introduced by Peng et al. (2021). We observe that our proposed approach achieves slightly better performance than Linear Transformer. It is important to note that AReLiT is more computationally efficient that Linear Transformer as it does not use a matrix as a recurrent state, and does not require calculating outer products to calculate the updates to the recurrent state.

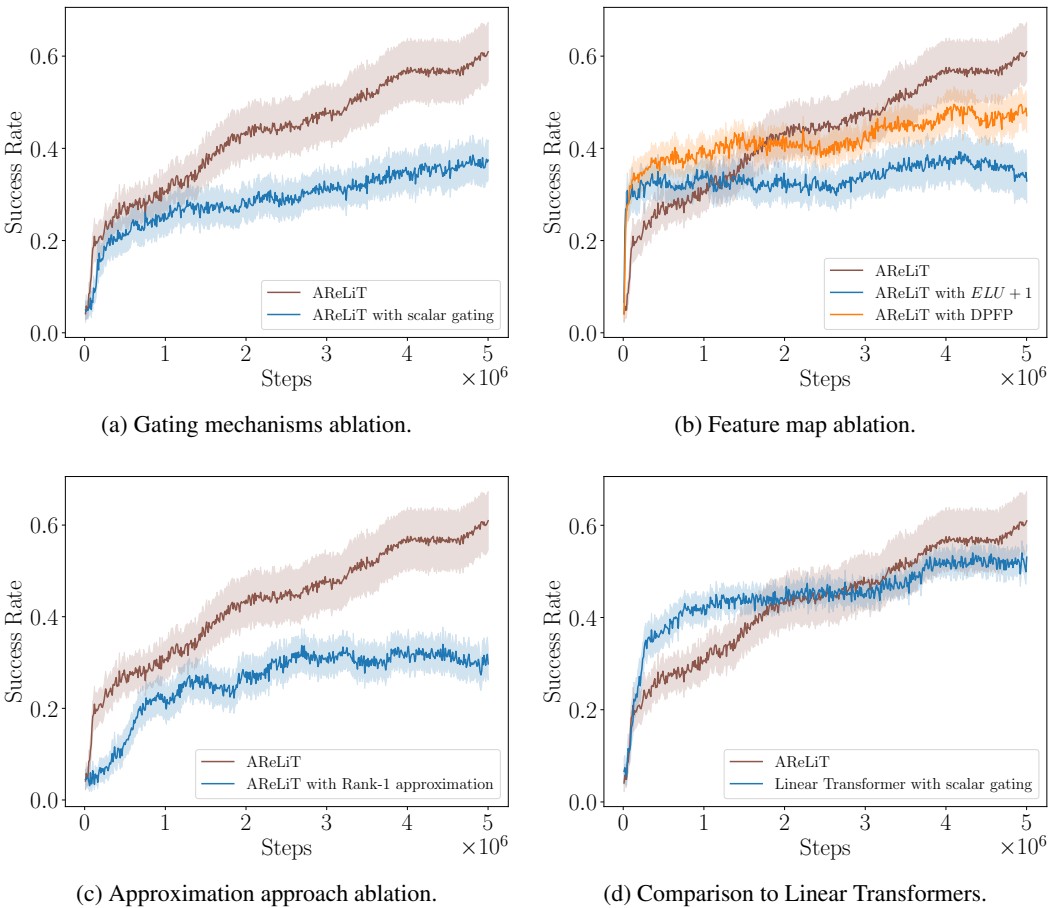

(a) Gating mechanisms ablation.

(b) Feature map ablation.

(c) Approximation approach ablation.

(d) Comparison to Linear Transformers.

Figure 13: Ablations for each of three proposed components in AReLiT. Results are over 50 seeds and the shaded region represents the standard error.

