# OpenReview forum: "Recurrent Linear Transformers"
_ICLR.cc/2024/Conference — Submitted to ICLR 2024_

### Official Review · Reviewer_gTNz · 2023-10-27

**Soundness:** 2 fair
**Presentation:** 2 fair
**Contribution:** 2 fair
**Rating:** 3
**Confidence:** 3

**Summary:**

The paper proposes a mechanism of positing transformer-like systems in the recurrent domain to alleviate the quadratic cost of inference in transformers with a linear complexity, as is observed in typical recurrent systems like RNNs. Prior work already achieves this by replacing the softmax operation of transformers with a kernel similarity mapping, but suffers from having to store a matrix as its state space, as well as an unconstrained addition of positive values which would lead to numerical instability. The authors augment this approach with a gating mechanism as well as an approximation to the kronecker delta function to alleviate both the problems, and evaluate their proposed mechanism in reinforcement learning regime where transformers cannot be naively used owing to large number of steps within each episode as well as the need for faster inference to enable fast data collection. The proposed mechanism, ReLiT and AReLiT show improved performance in resource constrained settings and can be seen as relevant substitutions in such an environment.

**Strengths:**

- The paper is well written and clearly demonstrates the benefits of the proposed approach when faced with a memory / compute restriction.
- The proposed mechanism (AReLiT) alleviates the need for storing a matrix in its recurrent state, thereby providing further savings in memory utilization.

**Weaknesses:**

- The approach in recurrent linear transformers (Katharopoulos et. al 2020) need not add just positive values at each iteration. In particular, it relies on a notion of similarity through kernel functions, where the kernel itself is positive but the mapping $\phi$ need not be. While Katharopoulos et. al do use a mapping based on positive values, it can simply be extended to setups where $\phi$ can also map to negative values; and hence this does not seem to be a limitation of their framework.
- The motivation behind a learnable feature map for self attention is not clear. Why not just do a parameterized $\phi$ represented through a small neural network, as opposed to the complex scheme of computing outer products for queries and keys? What is the motivation behind doing so?
- It would be useful if the authors did ablations for each of their additions, not only about how much the addition contributes to the performance but also what are some of the other candidate options. For example, what is the impact of using the outer product based query-key computation or using a normal neural network based approach without outer products.
- Given that an important benefit of the proposed approach is the savings obtained on time complexity, it would be nice to have some results highlighting the performance obtained by the different methods across wall clock time to really see the advantage of the method.
- There are also baselines that are missing in a number of experiments. None of the experiments benchmark against Katharopoulos et. al and there are also baselines missing in Figure 2.
- There are a few typos, eg. “In the RL” → In RL, “addresses” → address in Section 3, etc.

**Questions:**

The authors claim that a positive feature map is required to ensure that the similarity scores produced by the underlying kernel function are positive. Technically a kernel function only requires mapping to some vector space where dot-product is defined, and thus defining a kernel function does not require positive feature maps. Could the authors clarify this?

---

> ### Author Response · Authors · 2023-11-19
> **Response to Reviewer gTNz: 1/2**
>
> Dear reviewer gTNz, please see the general response to all reviewers about experiments and our focus on the RL setting. Thank you for carefully reading our paper and giving thoughtful comments and suggestions. We will respond to the key points of misunderstanding and answer your questions below in a point by point fashion.
>
> > The authors claim that a positive feature map is required… Could the authors clarify this?
>
> We agree that it is possible to have a non-positive feature map if the underlying kernel function produces non-negative similarity. We assumed that the feature map is positive, as done by Katharopoulos et. al (2020). We chose to follow this assumption for the following reasons:
> A positive feature map is a simple way to produce positive kernel scores and has been a common assumption in previous extensions to the linear transformer paper (Peng et al., 2021; Schlag et al. 2021)
> A positive feature map also ensures that the normalization vector $\mathbf{s}_t$ at each timestep $t$ is non-zero. This is necessary as calculating the attention vector $\mathbf{a}_t$ requires division by  $\mathbf{s}_t$.  $\mathbf{s}_t$ is defined as the sum of the query vectors. The assumption that the query vectors are positive is necessary, because otherwise the resultant sum $\mathbf{s}_t$ could become zero, causing the attention vector to be infinite.
> We will modify the main text to expand on these points. An alternative feature map could be designed such that it (1) doesn’t have the above normalization issue and (2) generates positive similarity scores; we did not explore this in this work and consider it in future work.
>
> > The motivation behind a learnable feature map for self attention is not clear. … What is the motivation behind doing so?
>
> The motivation behind a learnable feature map is to be able construct adaptable feature maps by learning them directly from the data. Our proposed feature map uses learnable parameters and an outer product operation to generate an expansive feature map. By introducing learnable parameters we are able to learn feature maps which are more suited to the data which they are trained on. This is in contrast to expansive feature maps in the existing literature (Peng et. al, 2021), which uses random vectors to generate an expansive feature map, and DPFP (Schlag et. al 2021) which assumes that the expansive features must be orthogonal.
>
> > Why not just do a parameterized represented through a small neural network, as opposed to the complex scheme of computing outer products for queries and keys?
>
> Using a neural network to generate a large feature map is interesting, but it will require more parameters than the proposed outer product-based mechanism. For instance, for an input vector of dimension $d$, consider that we want to produce an output feature of dimension $d\eta$. For concreteness lets assume $\eta$ is equal to 4 and $d$ is 64. Using a neural network to produce this feature would require $d^2\eta=16384$ parameters. On the other hand, our proposed outer-product-based rule requires $d\eta + d^2=4352$. Decoupling the $\eta$ hyperparameter from the $d^2$ allows us to increase the projection dimension to a large value without scaling quadratically with $d$.
>
> > Given that an important benefit of the proposed approach is the savings obtained on time complexity, it would be nice to have some results highlighting the performance obtained by the different methods across wall clock time to really see the advantage of the method.
>
> The current manuscript has experiments that measure the wall clock time of our proposed approach when compared to transformers with softmax-based self-attention. We provide these results in Appendix J, Figure 12, where we compare the wall clock time/latency of forward pass using GTrXL and AReLiT. We find that in comparison to canonical self-attention, our proposed approach achieves significant savings in wall clock time with increased context or with increased sequence length. We also provide empirical evidence of the savings obtained by our approach by measuring the frames per second (FPS) in the Memory Maze experiments in Section 5.

---

> > ### Author Response · Authors · 2023-11-19
> > **Response to Reviewer gTNz: 2/2**
> >
> > > There are also baselines that are missing in a number of experiments. None of the experiments benchmark against Katharopoulos et. al and there are also baselines missing in Figure 2.
> >
> > We agree with the reviewer that a comparison with Katharopoulos et al. is important. Fortunately, we have a comparison of our approach with Katharopoulos et. al (2020) augmented with a scalar gating mechanism (Peng et al. 2021) in Appendix K, Figure 13 (d). We performed this comparison in the T-Maze task with a corridor length of 200 and found that our proposed approach outperformed Katharopoulos et. al (2020).
> > As for Figure 2, we used GRU as our baseline for this diagnostic task since Morad et al. (2022) reported it to be the best-performing architecture (when comparing nearly a dozen other approaches) on partially observable classical control tasks, even when compared to transformer approaches.

---

> > > ### Comment · Reviewer_gTNz · 2023-11-21
> > > **Official Comment by Reviewer gTNz**
> > >
> > > Thanks to the authors for providing the clarifications.
> > >
> > > - Is it possible to see the proposed method as another instance of Katharopoulos et. al but with a different kernel function?
> > > - It is still not clear to me why the authors want to use outer-products in Equations 10 and 11? Is it just to scale up dimensionality in a cheap manner?
> > > - I think it would still be useful to have the baselines set up for Figure 2.

---

> > > > ### Author Response · Authors · 2023-11-21
> > > > **Response to Reviewer gTNz**
> > > >
> > > > > Is it possible to see the proposed method as another instance of Katharopoulos et. al but with a different kernel function?
> > > >
> > > > That would be an overstatement. Apart from the proposed feature map, the proposed approach has two additional contributions which are crucial for performance (See Appendix K) and computational benefits (see Appendix A) over Katharopoulos et. al. The motivation behind these two contributions are as follows:
> > > > 1. An outer-product based gating mechanism to control the flow of past information in the Linear Transformer self-attention: Gating mechanisms allow for a more fine grained control of past information that needs to be deleted and new information that gets added. This is important to handle long contexts as the network can selectively retain past information.
> > > > 2. Approximation of recurrent state update: approximation of recurrent state is essential to reduce the computational complexity of applying the recurrent state update.
> > > >
> > > > In addition to the kernel function, the outer product-based gating mechanism gives the architecture flexibility to attend to information from the past, and the approximation gives us computational efficiency. The results in the main text and in the appendix K provide empirical evidence of the benefits of these 3 components and over Katharopoulos et. al.
> > > >
> > > > > It is still not clear to me why the authors want to use outer-products in Equations 10 and 11? Is it just to scale up dimensionality in a cheap manner?
> > > >
> > > > Precisely, as demonstrated with the NN example in our previous comment, to scale up to dimension of 256 using a input feature of dimension of 64, we would need 3.76 times fewer parameters with an outer product based approach. With the outer product based feature map, the number of parameters is $d\eta + d^2$. Here, $\eta$ is the factor of times you want to increase the input dimensionality $d$ to. The key point here is that the number of features increased only $d$ times if we were to increase $\eta$.
> > > >
> > > > > I think it would still be useful to have the baselines set up for Figure 2
> > > >
> > > > By missing baselines does the reviewer mean GTrXL? In the main text we provided justification for why a transformer result was not included in Figure 2. Previous work done by Morad et al. (2020) demonstrated that GRU outperforms transformers in partially observable classical control problems. However, we could add these results in camera ready version of the paper.

---

### Official Review · Reviewer_EZRU · 2023-10-28

**Soundness:** 3 good
**Presentation:** 2 fair
**Contribution:** 2 fair
**Rating:** 5
**Confidence:** 4

**Summary:**

The quadratic complexity on the sequence length characterizing the self-attention mechanism has led to a surge of works aiming at optimizing Transformers performances, mainly inspired by recurrent mechanisms. Building on the Linear Attention (Katharopoulos et al., 2020) and subsequent works (Peng et al., 2021) , this paper proposes two recurrent alternatives to self-attention. The former, **ReLit**, introduces a learned gating mechanism that decays each index of the recurrent state matrix  $C_t$, in addition to a learned feature map $\phi$ instead of a fixed one (as in Katharopoulos et al., 2020 or Peng et al., 2021).  The latter proposed model, **AReLit**,  is a low-rank approximation of the former,  introduced to optimize the space complexity. The authors carried on an experimental analysis focused on Reinforcement Learning (RL) tasks, investigating the model capability.

**Strengths:**

The paper is well written, and the clarity and structure of the first sections help the reader in understanding the context. The model is an incremental improvement that builds upon Linear Attention (Katharopoulos et al., 2020) and subsequent works, especially (Peng et al., 2021). I believe that the relation and inspiration from (Peng et al., 2021) should be better highlighted, given that such paper already introduced a (scalar) gating mechanism in the recurrence of *causal* Linear Transformers. However, the approach is original and the topic is significant to the community.

**Weaknesses:**

The authors describe the Linear Transformer [1] as a recurrent approach. I believe that the recurrent view of transformers is feasible only in the *causal* modality, i.e. when **causal masking** is performed (causal attention, also referred to as “decoder self-attention” or “autoregressive attention” -- see both [1, 2]). Moreover, the paper title refers to a general *Recurrent Linear Transformer* whereas the main contributions are an index-based gating mechanism and a learnable feature map (i.e., the recurrence was already available in the models they build upon) --  I believe the paper title could be adjusted to reflect this.

Despite the very promising empirical evaluation on RL tasks, I believe that the model investigation could be improved.
The authors proposed two variants to surpass the limitations of Linear Transformers [1,2]. Hence, I would expect an in-depth investigation  and comparison with respect to such models and other similar competitors [3]. Apart from an ablation study  (relegated in the supplementary section K), there are not other intuitions.
Direct comparison in terms of number of learnable parameters (additional parameters introduced in Section 3.1 and 3.2 seems a lot), execution times or memory footprint  (see Table 3 and Figure 2 of [2]) are needed.
Moreover, why did the authors choose to tackle RL tasks instead of standard benchmark devised for Transformers/Long range dependecies [4]? (or the tasks performed by  [1,2,3])



[1] Katharopoulos, Angelos, et al. "Transformers are rnns: Fast autoregressive transformers with linear attention." International conference on machine learning. PMLR, 2020.

[2] Peng, Hao, et al. "Random Feature Attention." International Conference on Learning Representations. 2022.

[3] Peng, Bo, et al. "RWKV: Reinventing RNNs for the Transformer Era." arXiv preprint arXiv:2305.13048 (2023).

[4] Tay, Yi, et al. "Long Range Arena: A Benchmark for Efficient Transformers." International Conference on Learning Representations. 2020.

**Questions:**

1) Section 3.1 and 3.2 seem to introduce a large amount of new learnable parameters.  Direct comparison in terms of number of learnable parameters, execution times or memory footprint (see Table 3 and Figure 2 of [2]) with respect to direct competitors [1,2,3] are needed (also, the tradeoff w.r.t performances).  The authors only refer to timings in the last paragraph of page 8 (and supplementary section J), but it seems that the considered competitor is a standard self-attention Transformer and not a recurrence-based one.

2) The authors proposed **ReLit** and **AReLit** in order to solve the issue of positive value addition in Linear Transformers recurrence (that could grow arbitrarily large). It could be interesting to better investigate this issue (also empirically -- or at least make the reader understand where the Linear Transformer fails) and how the proposed model solve it.

3) The relation with Peng et al. [2] should be better highlited in the paper text, given that a (scalar) gating mechanism was already introduced in such paper.

4) I would expect an in-depth investigation  and comparison with respect to direct competitors [1,2, 3]. Apart from an ablation study  (relegated in the supplementary section K), there are not other intuitions. The central contribution of the paper is an improvement with respect to [1,2], hence the improvements should be the main investigation of the experimental evaluation.

5)  Why did the authors choose to tackle RL tasks instead of standard benchmark devised for Transformers/Long range dependecies [4]? (or even the tasks performed by  [1,2,3])  What happens when the sequence scale is increased (ListOps)?



[1] Katharopoulos, Angelos, et al. "Transformers are rnns: Fast autoregressive transformers with linear attention." International conference on machine learning. PMLR, 2020.

[2] Peng, Hao, et al. "Random Feature Attention." International Conference on Learning Representations. 2022.

[3] Peng, Bo, et al. "RWKV: Reinventing RNNs for the Transformer Era." arXiv preprint arXiv:2305.13048 (2023).

[4] Tay, Yi, et al. "Long Range Arena: A Benchmark for Efficient Transformers." International Conference on Learning Representations. 2020.

---

> ### Author Response · Authors · 2023-11-19
> **Response to Reviewer EZRU**
>
> Dear reviewer EZRU, please see the general response to all reviewers about experiments and our focus on the RL setting. Thank you for carefully reading our paper and giving thoughtful comments and suggestions. We will respond to the key points of misunderstanding and answer your questions below in a point by point fashion.
>
> > The relation with Peng et al. [2] should be better highlighted in the paper text, given that a (scalar) gating mechanism was already introduced in such paper.
>
> We agree with the reviewer that a comparison with Peng et al. is important as their work relates to our gating mechanism. Fortunately, we already have such a comparison as part of the ablation experiments in Appendix K, figure 13 (a), where we evaluated the impact of each of our proposed contributions, including gating mechanisms. We changed paragraph 4 of section 5, diagnostic MDP, to highlight these comparisons. To summarize the relation to Peng et al., we found that  the learning decay parameters for each element of $\mathbf{C}$ (gating) are better than a scalar decay used in the Linear Transformer (Peng et al. 2021) and that our proposed approach outperforms Linear Transformer with scalar gating presented by Peng et al. (2021). We will update the paper to (1) more clearly highlight that this comparison is in the appendix, and (2) expand the discussion of Peng et al. [2].

---

> > ### Comment · Reviewer_EZRU · 2023-11-22
> >
> > I acknowledge the authors response and thank them for the clarification on my Q3.
> > Since the authors pointed out that
> > > "We will respond to the key points of misunderstanding and answer your questions below in a point by point fashion."
> >
> > I was waiting for explicit answers also to the other points I raised. I understood that the general response roughly answered Q4/Q5 of my Review,  but I could not grasp any clarification on:
> >
> >   > Q1. Section 3.1 and 3.2 seem to introduce a large amount of new learnable parameters. Direct comparison in terms of number of learnable parameters, execution times or memory footprint (see Table 3 and Figure 2 of [2]) with respect to direct competitors [1,2,3] are needed (also, the tradeoff w.r.t performances). The authors only refer to timings in the last paragraph of page 8 (and supplementary section J), but it seems that the considered competitor is a standard self-attention Transformer and not a recurrence-based one.
> >
> > > Q2. The authors proposed ReLit and AReLit in order to solve the issue of positive value addition in Linear Transformers recurrence (that could grow arbitrarily large). It could be interesting to better investigate this issue (also empirically -- or at least make the reader understand where the Linear Transformer fails) and how the proposed model solve it.
> >
> > I believe that Q1 constitutes an important point.

---

> > > ### Author Response · Authors · 2023-11-23
> > > **Response to Reviewer EZRU**
> > >
> > > > Section 3.1 and 3.2 seem to introduce a large amount of new learnable parameters.
> > >
> > > In fact, the additional learnable parameters we introduce are of the same size as the weight vectors used to generate the key and query vectors; thus they only marginally increase the number of learnable parameters. For instance, in Appendix J Figure 13 (d), we compare the performance of AReLiT with Linear Transformers. With the same configuration, we observe that Linear Transformer uses 1,718,928 parameters, whereas AReLiT uses 1,803,072 parameters (less than 5% increase). We will include these numbers in the camera ready version of the paper.
> > >
> > > > Direct comparison in terms of number of learnable parameters
> > >
> > > In Appendix J, Figure 13 (d) we provide comparison of AReLiT with Linear Transformer ([1], [2]). As mentioned in our previous comment, the introduced additional parameters only marginally increase the total number of parameters.
> > >
> > >
> > > > Direct comparison in terms of ... execution times or memory footprint (see Table 3 and Figure 2 of [2]) with respect to direct competitors [1,2] ... are needed (also, the tradeoff w.r.t performances).
> > >
> > > The computational complexity of Linear Transformer is of the same order as that of Linear Transformer (Appendix A, Table 1). AReLiT, however, is more computationally efficient than Linear Transformer due to the introduced approximation. In terms of number of operations in T-Maze, we observed that ReLiT uses 18 times more memory to store the recurrent state, and 3.13 times more operations than AReLiT for a single attention head with similar configuration. In the camera ready version of the paper, we will include comparison of AReLiT wrt to Linear Transformer/ReLiT  in terms of wall-clock time (as demonstrated in Appendix J Figure 12.
> > >
> > > > Direct comparison in … with respect to direct competitors [3]
> > >
> > > Our experiments focus on RL. We don't understand how RWKV could be applied in RL. Extending RWKV to RL could be a whole paper in and of itself. Does the reviewer have suggested baselines that would be applicable in RL?
> > >
> > >
> > > > Q2. The authors proposed ReLit and AReLit in order to solve the issue of positive value addition in Linear Transformers recurrence (that could grow arbitrarily large). It could be interesting to better investigate this issue (also empirically -- or at least make the reader understand where the Linear Transformer fails) and how the proposed model solve it.
> > >
> > > It is tricky to report negative results in a paper. Obviously, our claim did not come out of nowhere. We did run vanilla Linear Transformers in T-Maze, however, due to the issue of positive value addition, the gradients quickly blew up and became NaNs after a few iterations. We can include this result in the camera ready version of the paper, by plotting the magnitude of the gradients during the initial steps of training.
> > >
> > > Finally, in Appendix J, Figure 13 (d) we compare the performance of Linear Transformer with scalar gating mechanism with AReLiT and found it to be worse.

---

> > > > ### Comment · Reviewer_EZRU · 2023-11-23
> > > >
> > > > I thank the authors for the clarification on the number of learnable parameters, I believe it is very important to highlight this on the paper to better compare with competitors.
> > > >
> > > > I did not get the comment on
> > > >
> > > > > Our experiments focus on RL. We don't understand how RWKV could be applied in RL. Extending RWKV to RL could be a whole paper in and of itself. Does the reviewer have suggested baselines that would be applicable in RL?
> > > >
> > > > Firstly, my comment was not only solely about RWKV [3], but about
> > > >
> > > > > Direct comparison in terms of number of learnable parameters, execution times or memory footprint (see Table 3 and Figure 2 of [2]) with respect to direct competitors [1,2,3] are needed (also, the tradeoff w.r.t performances).
> > > >
> > > > The metrics regarding *execution times or memory footprint* can be obtained by simply performing a forward pass on a synthetic sequence.
> > > > Moreover, the main contribution of the paper is the proposal of a new architecture claimed to be more efficient and have a better representational capability with respect to competitors (I pointed to [1,2,3] since they are among the best known efficient transformers). Thus, I would expect the main focus of the paper to prove this claims. As highlighted by *all the other Reviewers*, just reporting as the main result performances on RL tasks is not enough, given that the setting chosen by the authors did not  allow to show comparisons with the aforementioned competitors in the setting they tackled.
> > > > I confirm my previous score.

---

### Official Review · Reviewer_o1NU · 2023-10-29

**Soundness:** 3 good
**Presentation:** 3 good
**Contribution:** 2 fair
**Rating:** 5
**Confidence:** 4

**Summary:**

This paper presents recurrent linear transformers -- an extension of Linear Transformer that involves the gating of state information. The authors posit that the linear transformers suffer due to (i) excessive positive additions while updating hidden states and (ii) sensitivity to the choice of kernel feature maps used to calculate the queries, keys, and values. To mitigate these mentioned shortcomings, the authors present ReLiT and its approximation called AReLiT. These use (i) a gating mechanism to limit the excess flow of state information and (ii) parameterized feature maps that replace fixed kernel feature maps. To evaluate the efficiency of ReLiT and AReLiT, long-horizon RL tasks are chosen where it is essential to remember past observations and actions to make an informed decision. Particularly, environments like T Maze, Noisy CartPole, MysteryPath, and Memory Maze are employed for empirical analysis.

**Strengths:**

The paper aims at improving well-established transformer architecture for processing longer-length inputs. Especially in the case of deep sequential decision-making, having a sequence encoder that can process longer-length inputs is one of the coveted tools. The paper attempts a very relevant problem.

While the authors describe an extension to linear transformers, they also suggest an approximate version that reduces the space complexity of the proposed improvement further, making it practically viable.

I like the choice of environments, especially the T-Maze mentioned, where the agent has to remember the context of decision-making from the beginning of the episode. Plus, the Noisy CartPole environment is apt for testing the contextual abilities of different networks as the network has to model velocity using only displacement vectors.

**Weaknesses:**

The **main issue** with the current draft is it lacks a comparison of ReLiT with the linear transformer. From the introduction and the way the whole work is motivated, I was expecting to see empirical validation of the issues in the previous linear transformers, namely, the excessive positive feedback overburdening the hidden states in some way or the other, and the effect of fixed kernel functions. However, the draft falls short of underscoring these issues empirically. Unless these issues are substantiated, the motivation behind the work does not feel convincing.

**Related Works**: Also, since the paper is about coming up with a better architecture for processing longer inputs, I feel the discussion on the efforts to increase context lengths in transformers should be included in the paper. For instance, authors could start with initial attempts like LongFormer. The Recurrent Memory Transformer (RMT) is another very relevant direction. Authors can qualitatively compare their approach to RMTs, too.

**Evaluation metrics**: My understanding of the work is that the recurrent linear transformers would allow us to (i) have lower time and space complexity compared to traditional transformers and (ii) have effective hidden state usage. I am unsure whether focusing more on the expected returns of the RL agents will sufficiently capture the computational effectiveness or utility of gating. One of the reasons to raise this up is the paper, from its beginning, does not try to solve RL using transformers as a specialty. There are a lot of recent works, such as Decision Transformers, which try to attempt to solve RL using transformers. The paper, however, keeps their mention to minimal, which I am okay with as I evaluate the approach from the architectural superiority of ReLit against other contenders, and not from the perspective of return maximization.

**The reported time and space complexities**: At the end of page 2, the paper describes the time complexity of the self-attention module in canonical transformers to be O(N d^2), which does not seem correct. I tried calculating the complexity on my own and also verified with a few other sources, too, and found the time complexity to be O(N^2 d + N d^2). I also found it hard to understand why Linear Transformer's self-attention complexity was independent of N.

**Questions:**

I have the following questions cum suggestions for the authors:
1. The paper would really benefit from the addition of a comparison with linear transformers.
2. Authors could revise the related works section by adding discussion on the aforementioned related directions.
3. Authors could look into ways of highlighting the usefulness of the gating mechanism described in ReLit.
4. How do different kernel functions affect the working of ReLiT or linear transformers in general? Can the authors please demonstrate an ablation study?
5. Can authors please provide a step-by-step derivation of how time and space complexities are derived for different transformer variants?

---

> ### Author Response · Authors · 2023-11-19
> **Response to Reviewer o1NU**
>
> Thank you for carefully reading our paper and giving thoughtful comments and suggestions. We will respond to the key points of misunderstanding and answer your questions below in a point by point fashion.
>
> > The paper would really benefit from the addition of a comparison with linear transformers
>
> We have already done this in Appendix K, Figure 13 (d), where we compared ReLiT with linear transformers and showed the effect of different kernel feature maps, gating mechanisms, and approximation approaches. We will update the main text of the paper to more clearly point the reader to this result in the appendix.
>
> > Authors could revise the related works section by adding discussion on the aforementioned related directions.
>
> We thank the reviewer for highlighting these great papers from the literature. We will update the related work section to cover previous approaches that address increasing the context length of transformers.
>
> > Authors could look into ways of highlighting the usefulness of the gating mechanism described in ReLit.
>
> We have already done this Appendix K, Figure 13 (a), where we compare the gating mechanism proposed in ReLiT with existing gating mechanisms proposed for the Linear Transformer architecture.  We find that learning decay parameters for each element of $\mathbf{C}$ (gating) is better than a scalar decay used in the Linear Transformer (Peng et al. 2021). We will update the main text of the paper to more clearly point the reader to this result in the appendix
>
> > How do different kernel functions affect the working of ReLiT or linear transformers in general? Can the authors please demonstrate an ablation study?
>
> We have already done this Appendix K, Figure 13 (b), where we compare the kernel function proposed in ReLiT with feature maps used in the existing literature. We found that our expansive feature map such as ours outperforms element-wise maps like ELU+1 and deterministic feature maps like DPFP (Schlag et al. 2021). We will update the main text of the paper to more clearly point the reader to this result in the appendix.
>
> > At the end of page 2, the paper describes the time complexity of the self-attention module in canonical transformers to be O(N d^2), which does not seem correct. I tried calculating the complexity on my own …and found the time complexity to be O(N^2 d + N d^2).
>
> I see why you thought the complexity would be O(N^2 d + N d^2), because you assumed that the computational complexity is calculated for processing an entire sequence of length N. We, however, report the computational complexity of self-attention for processing a single element in a sequence.  For a matrix $\mathbf{X}$, the inference cost is the cost of processing a single element $\mathbf{X}[i]$. For canonical self-attention, the cost of processing a single element depends linearly on the input sequence length N. The following is the computational complexity of the steps performed by canonical self-attention:
> 1. Calculate the key and value vectors for N timesteps, and query for a single timestep: O(N d^2)
> 2. Calculate the dot product of query of a single timestep with key vectors of previous timesteps, apply softmax, and multiply the resultant with the value vectors for N timesteps: O(N d)
> This results in a overall complexity of O(N d^2 + N d) = O(N d^2)
>
>
> > I also found it hard to understand why Linear Transformer's self-attention complexity was independent of N.
>
> Similar to the complexities reported for canonical transformers, we report the computational cost of processing a single element using the Linear Transformer architecture. The Linear Transformer algorithm presented in Algorithm 2 requires only a fixed dimension hidden state to generate the output at a given timestep. As such, the computational complexity of self-attention for processing a single element is independent of the input sequence length N.

---

> > ### Comment · Reviewer_o1NU · 2023-11-21
> >
> > Thank you for your response. Here are a few additional comments:
> >
> > - Figure 13 is one of the important figures that supports many ideas presented in the paper, so I request the authors to move it to the main text, if possible. If not, as authors agree to act upon, they need to be clearly referred to in the main text. Also, I find it hard to convince myself that these non-referred results are the motivation behind the whole work, given (1) the success rate of Linear Transformers is marginally behind AReLiT (Fig. 13d), (2)  The gating mechanism proposed in AReLiT is a need of the architecture from Fig. 13a without which it cannot outperform existing Linear Transformers and so lack any definitive qualitative advantage over scalar gating in terms of information propagation, (3) AReLiT with Rank-1 approximation is bound to perform sub-optimally (Fig. 13c), hence does not really add any weight on arguments presented in work.
> >
> > This makes me reiterate that the motivation behind the work is quite weak. Moreover, these Linear Transformer comparisons could have easily been made part of comparison plots against GTrXL, which the current draft totally avoids in the main text.
> >
> > - The distinction between different types of complexities used for measuring transformer performance could be made clearer in the main text, although this is a minor comment.
> >
> > In lieu of the above concerns, I keep my score unchanged.

---

### Official Review · Reviewer_zjLV · 2023-11-01

**Soundness:** 3 good
**Presentation:** 2 fair
**Contribution:** 2 fair
**Rating:** 6
**Confidence:** 3

**Summary:**

This paper was motivated by the heavy computational costs in vanilla Transformers and Linear Transformers in previous work. They provided two variants of Transformers, called recurrent linear Transformer and approximate recurrent linear transformer. They claim that with the first construction, the inference cost for each token is independent of the context length, and they used low-rank approximation in the second construction. The low-rank approximation is well theoretically motivated and is convincing.

They did experiments on RL problems which are known as computationally challenging to Transformers. They showed that in several RL environments, their models can compete with SOTA, but with 40-50 percent less inference costs. Generally speaking, this is a paper with obvious strengths and weakness. I vote for weak accept since despite some weakness, I think their idea is well motivated and the creative low-rank approximation is very worth investigating.

**Strengths:**

1. Their idea is very well motivated. They explain in detail where the inference costs in vanilla Transformers and Linear transformers arise.

2. The performance on some RL environments show a great reduction on inference costs while their average rewards are basically the same as the previous models.

3. The low-rank approximation in the approximate recurrent linear Transformer is well theoretically motivated. The theory seems good, and the approximation of Kronecker function by cosine functions are very creative and inspiring.

4. The paper is generally well written and easy to follow.

**Weaknesses:**

1. From the high level, I think the biggest issue is that the authors only did experiments in RL. Although they explained that this is because RL is particularly difficult for Transformers, this is not convincing enough. I still believe that the authors should include some experiments on language data / some other tasks which Transformer usually applies. The reason is, your idea of recurrent linear TF is not motivated by dealing RL problem with Transformers, and Transformers were not originally invented for RL problems, so it is a little bit strange to only conduct experiments on RL.

2. The motivation part can be more precise and some of your motivation is not really important, from my perspective. In the third paragraph of the introduction section, you propose three motivations, while I think your methods do not really address the second one (the kernel function one). The authors claimed that the Transformers of linear Transformers have a manually-designed kernel function (softmax / Gaussian), but in your ReLiT model, although there are some learnable parameters in your kernel functions, the kernel function class is still manually-designed and fixed. Moreover, the kernel function class in ReLiT is not rich enough to make Transformers learn the 'globally optimal' kernel. You can only expect that the Transformers learn the 'optimal' kernel in the kernel class you specify.

From my perspective, I think the tunable kernel function may not be the main reason for the experimental success of ReLiT on RL problems. The main advantage of your model is the low inference costs introduced by the low-rank approximation, and this learnable kernel is kind of far from your main contribution. This will make your paper more distracted.

I am not sure how the authors think. My suggestion is, if you want to claim that the learnable parametric kernel is important and the kernel class you specify is good, you may need to do some ablation experiments. For example, what happen is you replace the softmax or fixed kernel function with this learnable kernel in the standard /linear Transformers?

3. In the last paragraph in the section 2, you claim 'The Linear Transformer’s self-attention has a context-independent inference cost, unlike the canonical self-attention mechanism', which I feel it kind of unfair. Although the inference cost for each token in linear Transformer is independent of the context length, the inference cost of a token matrix (a sequence) does depend on that, right? In sec 2.1, you showed the costs for standard TF for inferring a token matrix while in sec2.2, you showed the costs for one token. I believe you may modify your claim here.

**Questions:**

/

**Details Of Ethics Concerns:**

/

---

> ### Author Response · Authors · 2023-11-19
> **Response to Reviewer zjLV: 1/2**
>
> Thank you for carefully reading our paper and giving thoughtful comments and suggestions. We will respond to the key points of confusion and answer your questions below in a point by point fashion.
>
> In addition, please see the general response to all reviewers about experiments and our focus on the RL setting.
>
> > The motivation for the work is unclear
>
> The motivation behind  each of the three proposed contributions are as follows:
> 1. An outer-product based gating mechanism to control the flow of past information in the Linear Transformer self-attention: Gating mechanisms allow for a more fine grained control of past information that needs to be deleted and new information that gets added. This is important to handle long contexts as the network can selectively retain past information.
> 2. An outer-product based parameterized expansive kernel feature map: Using outer-product to generate an expansive feature map allows us to have a large feature map output dimension. Having a large feature map output dimension is essential as it correlates to the memory capacity (see Schlag et. 2021). Using a parameterized feature map allows us to learn the feature map directly from the data.
> 3. Approximation of recurrent state update: approximation of recurrent state is essential to reduce the computational complexity of applying the recurrent state update.
>
> Taken together, these three components are essential for constructing a transformer architecture that can be updated online in a streaming fashion, which requires it to be computationally efficient without, of course, losing expressiveness. The outer product-based gating mechanism gives the architecture flexibility to attend to information from the past, the expansive kernel map gives us flexibility on the representation for learning it from data, and the approximation gives us computational efficiency. The results in the main text and in the appendix (highlighted below) provide empirical evidence of the benefits of these 3 components.
>
> > The motivation part can be more precise and some of your motivation is not really important, from my perspective….while I think your methods do not really address the second one (the kernel function one).
>
> As always motivation can be sharpened. Existing kernel functions which use element-wise (Katharopoulos et al., 2020) or randomized expansive (Peng et al., 2021) feature maps have been shown to have limited memory capacity (Schlag et al., 2021). Schlag et al. (2021) showed that expansive feature maps outperform element-wise feature maps such as ELU+1. They also showed that deterministic feature maps have higher memory capacity than randomized ones. Following the results in Schlag et al. (2021) our motivation was to design a feature map that is deterministic and generates an expansive representation of the input. We calculate an expansive representation of the input vector using outer products. As an added benefit, our proposed feature map is parameterized to enable learning more expressive expansive representations of the input.
>
> > The authors claimed that the Transformers of linear Transformers have a manually-designed kernel function (softmax / Gaussian), but in your ReLiT model, although there are some learnable parameters in your kernel functions, the kernel function class is still manually-designed and fixed.
>
> We did not mean to imply that manually designed kernel functions are bad, more so we should have emphasized that our kernel functions are better because our proposed kernel function uses learnable parameters to learn an expansive feature map, instead of using a static function. It is deterministic similar to DPFP (Schlag et al., 2021), but the learnable parameters allow it to learn a more fine tuned feature map from the training data.
>
> > Moreover, the kernel function class in ReLiT is not rich enough to make Transformers learn the 'globally optimal' kernel. You can only expect that the Transformers learn the 'optimal' kernel in the kernel class you specify.
>
> We are not sure what you exactly mean by a ‘globally optimal kernel’? Can you clarify what you mean?
>
> > I think the tunable kernel function may not be the main reason for the experimental success of ReLiT on RL problems. The main advantage of your model is the low inference costs introduced by the low-rank approximation, and this learnable kernel is kind of far from your main contribution.
>
> We agree that a major advantage of our method is the computation win from the low-rank approximation. However, we do not agree the learned kernel is not important as our feature map ablation experiment in Appendix K, Figure 13 (b) shows that the proposed feature map outperforms deterministic expansive feature map approach DPFP and element-wise feature map ELU+1.

---

> > ### Author Response · Authors · 2023-11-19
> > **Response to Reviewer zjLV: 2/2**
> >
> > > ...if you want to claim that the learnable parametric kernel is important and the kernel class you specify is good, you may need to do some ablation experiments. For example, what happen is you replace the softmax or fixed kernel function with this learnable kernel in the standard /linear Transformers?
> >
> > We actually included some of these ablations in the appendix of the submitted paper.  To highlight the advantage of our proposed feature map compared to existing feature maps, we present an ablation conducted in corridor length 200 in the T-Maze task. In Appendix K, Figure 13 (b), we show that replacing the proposed feature map in AReLiT with previously used feature maps such as ELU and DPFP caused degradation in performance.
> >
> > > Although the inference cost for each token in linear Transformer is independent of the context length, the inference cost of a token matrix (a sequence) does depend on that, right?
> >
> > Yes, in fact the computational cost of linear transformer and its derivative approaches are independent of the context length only when processing a single token, and not a token matrix.
> >
> > > In sec 2.1, you showed the costs for standard TF for inferring a token matrix while in sec2.2, you showed the costs for one token. I believe you may modify your claim here.
> >
> > We do not need to modify our claim because all the computational costs presented in our paper regarding our approach, the canonical transformer and linear transformers are in terms of the inference cost.   Let’s be clear about what we mean by inference cost: the cost of processing a single element in a sequence. We report all complexities under this definition of inference cost. In Sec 2.1, we report the computational cost of processing a single element using the canonical transformer, which depends linearly on the input sequence length N, as canonical self-attention requires past input to be provided to generate the output representation at a given timestep. In Sec 2.2, we report the computational cost of processing a single element using ReLiT, which in contrast to canonical transformers, doesn’t depend on having the full sequence.

---

### Author Response · Authors · 2023-11-19
**General response to all reviewers**

A common concern expressed by reviewers is that our evaluation is performed in RL tasks instead of the standard tasks used by the community when evaluating transformer architectures.  That was by design. In fact, even the base algorithm we chose, GTrXL, was influenced by the need to have RL agents that can successfully do non-linear function approximation with transformers.

We wanted to start by highlighting that we want to have the ability to thoroughly analyze our approach to be able to provide the ablations in Appendix K. Given the computational resources we have available to us, we wouldn’t be able to do something similar in bigger tasks, but we were in T-Maze. Importantly, RL problems are also more configurable than a dataset (as in the T-Maze in which we can control how long an agent needs to remember a piece of information). In fact, originally we did start this work evaluating our algorithms on next word prediction and the results were similarly promising, but we made the shift to RL during the project. We are currently rerunning that experiment and we will post the results and update the paper as soon as those come back. Unfortunately, due to the time constraints and the computational resources we have access to, we cannot provide results on other language tasks until camera ready.

When scaling up our experiments, we continued to use online RL tasks to emphasize that the architecture we are proposing is applicable to such a setting, something that is not true for most results. Naturally, as we wrote in our paper, RL tasks also emphasize the need for a faster inference time and the need for keeping a much longer context than it is usual in other settings. Finally, we consider the return an agent obtains a much clearer metric of performance, with semantics that are much better understood.

---

### Meta-Review · Area_Chair_9QPE · 2023-12-11

**Metareview:**

The submission follows in an interesting line of work on incorporating recurrent processing in transformer models. However, given the limited evaluation setting and the small increment from prior work, this paper is not yet ready for publication at the conference.

**Justification For Why Not Higher Score:**

limited evaluation of a work that builds on other studies puts into question the broader impact

**Justification For Why Not Lower Score:**

N/A

---

### Decision · Program_Chairs · 2024-01-16

Reject